# Robust Federated Learning under Heterogeneous Data with Generalized Heavy-Ball Momentum

**Riccardo Zaccone**[*]
Politecnico di Torino
riccardo.zaccone@polito.it

**Sai Praneeth Karimireddy**
USC Viterbi School of Engineering
karimire@usc.edu

**Carlo Masone**
Politecnico di Torino
carlo.masone@polito.it

**Marco Ciccone**
Vector Institute
marco.ciccone@vectorinstitute.ai

## Abstract

Reliable machine learning requires robustness to *unreliable* and *heterogeneous* data, a challenge that is particularly acute in Federated Learning (FL). Standard optimization methods degrade under the combined effects of data heterogeneity and partial client participation, while existing momentum variants introduce biased updates that undermine reliability. We propose a novel *Generalized Heavy-Ball Momentum* (GHBM), a principled optimization method that eliminates this bias and provides convergence guarantees even under *unbounded heterogeneity* and *cyclic participation*. We further develop adaptive, communication-efficient variants that retain the efficiency of FEDAVG. Extensive experiments on vision and language benchmarks confirm that GHBM substantially improves robustness and reliability compared to state-of-the-art FL methods, particularly in large-scale settings with limited participation. These results establish GHBM as a reliable foundation for distributed learning in environments with imperfect data [2].

## 1 Introduction

Machine learning systems deployed in the real world must contend with **unreliable data sources**: information is often **heterogeneous across users, incomplete due to limited participation, and subject to distribution shift or noise**. Ensuring **reliable training** under such conditions is a key challenge for the broader deployment of trustworthy ML.

Federated Learning (FL) [McMahan et al., 2017] provides a natural framework for this setting, enabling a central server to train a shared model by orchestrating local training across decentralized clients without requiring raw data sharing. While this setup offers important privacy advantages, it also introduces severe **reliability challenges**. Local datasets reflect unique characteristics of each client, and optimization restricted to personal data causes **statistical heterogeneity**, which in turn leads to *client drift* when synchronization is infrequent [Karimireddy et al., 2020]. These issues become even more acute under *partial client participation*, where only a fraction of clients contribute updates at each round.

A variety of methods have been proposed to mitigate heterogeneity, such as control-variates in SCAFFOLD [Karimireddy et al., 2020] or ADMM-based alignment in FEDDYN [Acar et al., 2021]. While theoretically well-motivated, these approaches often lack robustness in practice, exhibiting **instability or slow convergence under extreme heterogeneity, sparse participation, or large-scale**

---

[*]Corresponding author
[2]Code is available at `https://github.com/RickZack/GHBM`

39th Conference on Neural Information Processing Systems (NeurIPS 2025) Workshop: Reliable ML from Unreliable Data.

**deployments** [Varno et al., 2022]. Momentum methods, which are widely effective in centralized training, have also been adapted to FL [Hsu et al., 2019, Ozfatura et al., 2021, Xu et al., 2021, Karimireddy et al., 2021]. However, while their advantages are established under full participation [Cheng et al., 2024], we show that they become biased and unreliable in the presence of heterogeneity and partial participation, preventing them from correcting client drift effectively.

**Contributions.**   In this work, we address these limitations and advance the reliability of FL under unreliable data sources:

- We propose a novel momentum formulation, *Generalized Heavy-Ball Momentum* (GHBM), which eliminates the bias of classical momentum and yields communication-efficient variants that remain robust in the presence of extreme heterogeneity.
- We establish **non-convex** convergence guarantees for GHBM under cyclic partial participation, showing reliability even under unbounded data heterogeneity.
- Through extensive experiments on vision and language tasks, we demonstrate that existing methods break down in unreliable data regimes, while GHBM consistently achieves faster convergence and higher model quality, establishing it as a reliable foundation for federated learning in large-scale, imperfect data environments.

## 2   Related works

**The Problem of Statistical Heterogeneity.**   The detrimental effects of non-iid data in FL were first observed by [Zhao et al., 2018], who proposed mitigating performance loss by broadcasting a small portion of public data to reduce the divergence between clients' distributions. Recognizing weight divergence as a source of performance loss, FEDPROX [Li et al., 2020] adds a regularization term to penalize divergence from the global model. Other works [Kopparapu and Lin, 2020, Zaccone et al., 2022, Zeng et al., 2022, Caldarola et al., 2021] explored grouping clients based on their data distribution to mitigate the challenges of aggregating divergent models.

**SVRG and ADMM in FL.**   Stochastic variance reduction techniques have been applied in FL [Chen et al., 2021, Li et al., 2019] with SCAFFOLD Karimireddy et al. [2020] providing for the first time convergence guarantees for arbitrarily heterogeneous data. Besides doubling the communication to exchange the control variates, and it has been experimentally proved not robust enough to handle large-scale scenarios akin to cross-device FL [Reddi et al., 2021, Karimireddy et al., 2021]. Similarly, SCAFFOLD-M [Cheng et al., 2024] integrates classical momentum into SCAFFOLD. However, it still relies on variance reduction to tackle heterogeneity, inheriting and the same limitations of SCAFFOLD, as the ineffectiveness of variance reduction in deep learning [Defazio and Bottou, 2019]. Other methods are based on the Alternating Direction Method of Multipliers [Chen et al., 2022, Gong et al., 2022, Wang et al., 2022]. In particular, FEDDYN[Acar et al., 2021] dynamically modifies the loss function such that the model parameters converge to stationary points of the global empirical loss. Besides enjoying similar theoretical guarantees than SCAFFOLD, in practical cases it has displayed problems in dealing with pathological non-iid settings [Varno et al., 2022].

**Use of Momentum as Local Correction.**   As a first attempt, Hsu et al. [2019] adopted momentum at server-side to reduce the impact of heterogeneity. However, it has been proven of limited effectiveness under high heterogeneity, because the drift happens at the client level. This motivated later approaches that apply server momentum at each local step [Ozfatura et al., 2021, Xu et al., 2021], and the more general approach by Karimireddy et al. [2021] to adapt any centralized optimizer to cross-device FL. Rather differently from previous works, we propose a novel formulation of momentum specifically designed to take incorporate the descent information of clients selected at past $\tau$ rounds, which generalizes the classical heavy-ball [Polyak, 1964]. Most notably, we prove that our GHBM algorithm converges under arbitrary heterogeneity in cyclic partial participation - the first momentum method achieving this result without relying on other mechanisms like variance reduction. Extended discussion of related works is deferred to Appendix A.1.

## 3   Method

### 3.1   Setup

In FL a server and a set $\mathcal{S}$ of clients collaboratively solve a learning problem. At each round $t \in [T]$, a fraction of $C \in (0, 1]$ clients $\mathcal{S}^t \subseteq \mathcal{S}$ is selected. Each client $i \in \mathcal{S}^t$ receives the server model $\theta_i^{t,0} \equiv \theta^{t-1}$, and performs $J$ local optimization steps, using stochastic gradients $\tilde{g}_i^{t,j}$ evaluated on

local parameters $\theta_i^{t,j-1}$ and a batch $d_{i,j}$, sampled from its local dataset $\mathcal{D}_i$. In this work we formalize the learning objective as a finite-sum optimization problem, where each function is the local clients' loss function with only access to that client's stochastic samples:

$$\arg\min_{\theta \in \mathbb{R}^d} \left[ f(\theta) := \frac{1}{|\mathcal{S}|} \sum_{i \in \mathcal{S}} \left( f_i(\theta) := \mathbb{E}_{d_i \sim \mathcal{D}_i} [f_i(\theta; d_i)] \right) \right] \tag{1}$$

### 3.2 Generalized Heavy-Ball Momentum (GHBM)

In this section, we introduce our novel formulation for momentum, which we call *Generalized Heavy-Ball Momentum* (GHBM). First, we recall that classical momentum consists of a moving average of past gradients, and it is commonly expressed as in Eq. (2), which can be equivalently expressed in a version commonly referred to as *heavy-ball momentum* in Eq. (3) (see Lemma B.1):

**HEAVY-BALL MOMENTUM (HBM)**

$$\tilde{m}^t \leftarrow \beta \tilde{m}^{t-1} + \tilde{g}^t(\theta^{t-1}; \mathcal{D}^t) \tag{2} \qquad \tilde{m}^t \leftarrow (\theta^{t-1} - \theta^{t-2}) \tag{3}$$
$$\theta^t \leftarrow \theta^{t-1} - \eta \tilde{m}^t \qquad\qquad\qquad \theta^t \leftarrow \theta^{t-1} - \eta \tilde{g}^t(\theta^{t-1}; \mathcal{D}^t) + \beta \tilde{m}^t$$

**Overcoming the limitations of classical momentum in FL.** The gradient referred to above as $\tilde{g}^t$ is built from updates of clients $i \in \mathcal{S}^t$, which are usually a small portion of all the clients participating in the training. Consequently, at each round the momentum is updated using a direction biased towards the distribution of clients selected in that round. The core idea behind GHBM is updating the momentum term at each round with a reliable estimate of the gradient w.r.t. the global data distribution of all clients, *i.e.* using the average gradient of clients selected in the last $\tau$ rounds at current parameters $\theta^{t-1}$, as in Eq. (4), and set $\tau$ such that this condition is realized.

**DESIRED MOMENTUM UPDATE** | **PRACTICAL MOMENTUM UPDATE**

$$\tilde{m}^t \leftarrow \beta \tilde{m}^{t-1} + \frac{1}{\tau} \sum_{k=t-\tau+1}^{t} \tilde{g}^k(\theta^{t-1}; \mathcal{D}^k) \qquad\qquad \tilde{m}^t \leftarrow \beta \tilde{m}^{t-1} + \frac{1}{\tau} \sum_{k=t-\tau+1}^{t} \tilde{g}^k(\theta^{k-1}; \mathcal{D}^k)$$

$$\tag{4} \qquad\qquad\qquad\qquad\qquad\qquad\qquad\qquad\qquad \tag{5}$$

Eq. (4) cannot be implemented in partial participation, but can be approximated by reusing old gradients calculated at parameters $\theta^{k-1}$, as shown in Eq. (5). The introduced *lag* due to staleness which can be controlled in theory and that ultimately we show to be greatly compensated by the achieve reduction in heterogeneity (see Fig. 6). With this idea in mind, our proposed formulation consists of calculating the momentum term as the decayed average of past $\tau$ momentum terms, instead of explicitly using the server pseudo-gradients at the last $\tau$ rounds, as shown in Eq. (6). This formulation is close to the update rule sketched in Eq. (5) and has the additional advantage of enjoying a heavy-ball form similar to Eq. (3) (see Lemma B.2), which will be useful for deriving communication-efficient FL algorithms:

**GENERALIZED HEAVY-BALL MOMENTUM (GHBM)**

$$\tilde{m}_\tau^t \leftarrow \frac{1}{\tau} \sum_{k=1}^{\tau} \beta \tilde{m}_\tau^{t-k} + \tilde{g}^t(\theta^{t-1}; \mathcal{D}^t) \quad (6) \qquad \tilde{m}_\tau^t \leftarrow \frac{1}{\tau} \left( \theta^{t-1} - \theta^{t-\tau-1} \right) \tag{7}$$
$$\theta^t \leftarrow \theta^{t-1} - \eta \tilde{m}_\tau^t \qquad\qquad\qquad\qquad \theta^t \leftarrow \theta^{t-1} - \eta \tilde{g}^t(\theta^{t-1}; \mathcal{D}^t) + \beta \tilde{m}_\tau^t$$

Trivially, GHBM with $\tau = 1$ recovers the classical momentum, hence it can be considered as a generalized formulation. The GHBM term is then embedded into local updates using the heavy-ball form shown in Eq. (7), leading to the following update rule:

$$\textbf{CLIENT STEP:} \qquad \theta_i^{t,j} \leftarrow \theta_i^{t,j-1} - \eta_l \tilde{g}_i^{t,j}(\theta_i^{t,j-1}; d_i^{t,j}) + \underbrace{\frac{\beta}{\tau J} \left( \theta^{t-1} - \theta^{t-\tau-1} \right)}_{\tau - \text{GHBM}} \tag{8}$$

**Discussion on $\tau$.** The $\tau$ hyperparameter in GHBM controls the number of server pseudo-gradients to average when estimating the update to the momentum term. Intuitively, when considering only the effect on heterogeneity reduction, the optimal value would be the one that provides the average over all clients *i.e.* $\tau = 1/C$, which is the inverse of the client participation rate. As we demonstrate, this is the key factor that allows GHBM to converge under arbitrary heterogeneity, achieving the same convergence rate in *cyclic partial participation* as methods based on classical momentum attain in *full participation* (see Sec. 4.1).

### 3.3 Communication Complexity of GHBM and Efficient Variants

GHBM requires the server to additionally send the momentum term $\tilde{m}_\tau^t$, which introduces a communication overhead of $1.5\times$ w.r.t. FEDAVG, as momentum is usually applied to all model parameters. However, this overhead can be avoided by exploiting the fact that GHBM has an equivalent heavy-ball form, and noting that if clients participate cyclically, clients had already received the previous model $\theta^{t-\tau-1}$. This is still true on average under uniform client sampling, *i.e.*, calling $\tau_i$ the sampling period for client $i$, $\mathbb{E}[\tau_i] = \tau = 1/C$. In practice, the additional requirement on communication can be traded with persistent storage at the clients. In this algorithm, which we call **LOCALGHBM**, $\tau_i$ is adaptive and determined stochastically by client participation. The space complexity is constant in the size of model parameters for the clients and the communication complexity is the same as FEDAVG. We empirically found that performance can be further improved by considering $\theta_{i,j}^t$ instead of $\theta^{t-1}$ and $\theta_i^{t-\tau_i}$ instead of $\theta^{t-\tau_i-1}$ when calculating $\tilde{m}_{\tau_i}^t$. This final communication-efficient update rule is named **FEDHBM**. Although based on the same principle, our algorithms are suitable for different scenarios, which we discuss more in detail in Appendix A.4.

## 4 Theoretical Discussion

Our results rely on notions of stochastic gradient with bounded variance (4.1) and the smoothness of the clients' objective functions (4.2), which are common in deep learning. We introduce the additional assumption that clients participate following a cyclic pattern, which serves only as a technical detail needed to deterministically quantify the contributions of the clients to the GHBM momentum term (see discussion in Appendix A.3). Finally, Assumption 4.3 is introduced to facilitate comparisons with other algorithms that require it, while it not used in the proof of our Thm. 4.6.

**Assumption 4.1** (Unbiasedness and bounded variance of stochastic gradient)**.**

$$\mathbb{E}_{d_i \sim \mathcal{D}_i} [\tilde{g}_i(\theta; d_i)] = g_i(\theta; \mathcal{D}_i)$$

$$\mathbb{E}_{d_i \sim \mathcal{D}_i} \left[ \|\tilde{g}_i(\theta; d_i) - g_i(\theta; \mathcal{D}_i)\|^2 \right] \leq \sigma^2$$

**Assumption 4.2** (Smoothness of client's objectives)**.** Let it be a constant $L > 0$, then for any $i$, $\theta_1$, $\theta_2$ the following holds:

$$\|g_i(\theta_1) - g_i(\theta_2)\|^2 \leq L^2 \|\theta_1 - \theta_2\|^2$$

**Assumption 4.3** (Bounded Gradient Dissimilarity)**.** There exist a constant $G \geq 0$ such that, $\forall i$, $\theta$:

$$\frac{1}{|\mathcal{S}|} \sum_{i=1}^{|\mathcal{S}|} \|g_i(\theta) - g(\theta)\|^2 \leq G^2$$

**Assumption 4.4** (Cyclic Participation)**.** Let $\mathcal{S}^t$ be the set of clients sampled at any round $t$. A sampling strategy is *"cyclic"* with period $p = 1/C$ if:

$$\mathcal{S}^t = \mathcal{S}^{t-p} \qquad \forall\, t > p \quad \wedge$$
$$\mathcal{S}^k \cap \mathcal{S}^t = \varnothing \qquad \forall\, k \in (t-p, t)$$

*Remark* 4.5. While Thm. 4.6 relies on Assumption 4.4, **cyclic participation is not enforced in the experiments**, where we select clients randomly and uniformly. For a more comprehensive discussion on the role of the cyclic participation assumption in our work, we refer the reader to Appendix A.3.

### 4.1 Convergence Guarantees

We provide the convergence rate for GHBM for ***non-convex*** functions in (cyclic) partial participation. Comparison with recent related algorithms in Tab. 4. The proof is deferred to Appendix B.

**Theorem 4.6.** *Under Assumptions 4.1, 4.2 and 4.4, if we take $\tilde{m}_\tau^0 = 0$, and $\beta$, $\eta$ and $\eta_l$ as in Eq. (119), then* GHBM *with $\tau = 1/C$ converges as:*

$$\frac{1}{T} \sum_{t=1}^T \mathbb{E}\left[ \|\nabla f(\theta^{t-1})\|^2 \right] \lesssim \frac{L\Delta}{T} + \sqrt{\frac{L\Delta\sigma^2}{|\mathcal{S}|JT}}$$

*where $\Delta := f(\theta^0) - \min_\theta f(\theta)$, $\eta_l \leq \mathcal{O}(1/\sqrt{\tau})$ (see Eq. (119)) and $\lesssim$ absorbs numeric constants.*

**Discussion.** The rate of GHBM shows two major improvements: (i) it does not rely on the BGD assumption (4.3) and (ii) the dominant term on the right-hand side (RHS) scales with the size of all client population $|\mathcal{S}|$, instead of the clients selected in a single round $|\mathcal{S}|C$, thanks to incorporating old gradients. Further connection with centralized optimization and discussion on he use of cyclic participation are deferred to Appendices A.2 and A.3.

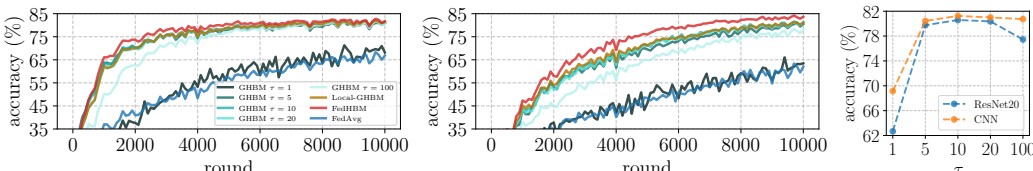

Figure 2: **GHBM effectively counteracts the effects of heterogeneity:** our momentum formulation ($\tau > 1$) is crucial for superior performance , with an optimal value $\tau = 1/C = 10$, as predicted in theory. Results on CIFAR-10 with CNN (left) and RESNET-20 (right), under worst-case heterogeneity.

**Comparison with FedCM.** The best-known rate for FEDCM in partial participation relies bounded gradient dissimilarity while in *full participation*, Cheng et al. [2024] proved that FEDCM converges under unbounded heterogeneity (see Tab. 4). We prove that GHBM can achieve the same convergence rate even in cyclic partial participation: indeed, as a validation, in Figure 1 we simulate a cyclic participation setting, comparing GHBM with FEDCM, both when selecting a subset of clients and when selecting them all. As it is shown, the curve of GHBM with $\tau$ as prescribed by Thm. 4.6 approaches the one of FEDCM in full participation.

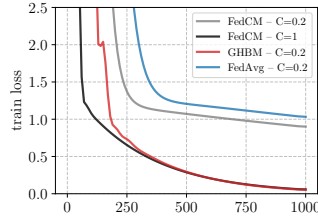

Figure 1: **Comparison between FEDCM and GHBM** in *cyclic participation* on a linear regression problem (see Appendix C.1 for details). GHBM with $\tau = 1/C$ in *cyclic participation* ($C = 0.2$) performs similarly as FEDCM in *full participation* ($C = 1$).

## 5 Experimental Results

**Scenarios, Datasets and Models.** For the controlled scenarios, we employ CIFAR-10/100 as computer vision tasks, with RESNET-20 and the same CNN similar to a LeNet-5 commonly used in FL works [Hsu et al., 2020], and SHAKESPEARE dataset as NLP task following [Reddi et al., 2021, Karimireddy et al., 2021]. For simulating settings akin to *cross-device* FL, we adopt the large-scale GLDV2 and INATURALIST datasets as CV tasks, with both a VIT-B\16 [Dosovitskiy et al., 2021] and a MOBILENETV2 [Sandler et al., 2018] pretrained on ImageNet, and STACKOVERFLOW dataset as NLP task. Further details on datasets, splits, models and hyperparameters are in Appendix C.

**Metrics and Experimental protocol.** We consider *final model quality*, as the average top-1 accuracy over the last 100 rounds of training (Tabs. 1 and 2), and *communication/computational efficiency*, evaluated by measuring the total amount of exchanged bytes (*i.e.* considering both the downlink/uplink communication) and the wall-clock time spent by an algorithm to reach the performance of FEDAVG (Tab. 3). **All the experiments are conducted under random uniform client sampling**.

### 5.1 The Effectiveness of GHBM Compared to Classical Momentum

In Fig. 2 we show the effectiveness of GHBM compared to classical momentum, which corresponds to selecting $\tau = 1$ in the update rule in Eq. (8), and simulate a scenario of extreme heterogeneity (*i.e.* $\alpha = 0$). Methods based on classical momentum [Xu et al., 2021, Ozfatura et al., 2021] fail to improve upon FEDAVG, while, in contrast, as $\tau$ increases, GHBM exhibits a significant enhancement in both convergence speed and final model quality. The optimal value of $\tau$ is experimentally determined to be $\tau \approx 1/C = 10$, with larger sub-optimal values only slightly affecting performance (rightmost plot).

### 5.2 Comparison with the State-of-art

**Results in Controlled Scenario.** We compare GHBM with the most common FL methods, and in particular with other momentum-based FL algorithms Results in Tab. 1 underscore that methods based on classical momentum fail at improving FEDAVG under high heterogeneity and partial participation, confirming the limitations outlined in Sec. 3.2. Conversely, our algorithms outperform FEDAVG with an impressive margin of $+20.6\%$ and $+14.4\%$ on RESNET-20 and CNN under worst-case heterogeneity, and consistently over less severe conditions (*i.e.* higher values of $\alpha$ in Fig. 3).

**Results in Real-world Large-scale Scenarios.** Extending the experimentation to settings characterized by extremely low client participation, we test both our GHBM with $\tau$ tuned via a grid-search and our adaptive FEDHBM, which exploits client participation to keep the same communication complexity of FEDAVG. As discussed in Sec. 3.2, under such extreme client participation patterns GHBM performs better because the trade-off between heterogeneity reduction and gradient lag is

explicitly tuned by the choice of the best performing $\tau$, while FEDHBM will likely adopt a suboptimal value. However, results in Tab. 2 show a stark improvement over the state-of-art for both our algorithms, indicating that the design principle of our momentum formulation is remarkably robust and provides effective improvement even when client participation is very low (*e.g.* $C \leq 1\%$).

**Communication Efficiency.** Results in Tab. 3 reveal that our proposed algorithms show faster convergence and higher final model quality, with an average saving of respectively $+55.9\%$ and $+61.5\%$. In particular, in settings with extremely low client participation (*e.g.* GLDv2), GHBM is more suitable for best accuracy, while FEDHBM is the best at lowering the communication cost.

Table 1: **Comparison with state-of-art in controlled setting** (acc@10$k$-20$k$ rounds for RESNET-20/CNN). NON-IID ($\alpha = 0$) and IID ($\alpha = 10.000$). Best result in **bold**, second best underlined. ✗ indicates non-convergence.

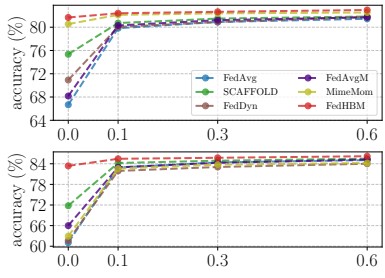

Figure 3: **Final model quality at different values of $\alpha$** (lower $\alpha \rightarrow$ higher heterogeneity) on CIFAR-10, with CNN (top) and RESNET-20 (bottom).

| METHOD | CIFAR-100 (RESNET-20) | | CIFAR-100 (CNN) | | SHAKESPEARE | |
|---|---|---|---|---|---|---|
| | NON-IID | IID | NON-IID | IID | NON-IID | IID |
| FEDAVG | 24.7 ±1.2 | 58.6 ±0.4 | 38.3 ±0.3 | 49.7 ±0.2 | 47.3 ±0.1 | 47.1 ±0.2 |
| FEDPROX | 24.8 ±1.1 | 58.5 ±0.3 | 40.6 ±0.2 | 49.9 ±0.2 | 47.3 ±0.1 | 47.1 ±0.2 |
| SCAFFOLD | 30.7 ±1.3 | 58.0 ±0.6 | 45.5 ±0.1 | 49.4 ±0.4 | 50.2 ±0.1 | 50.1 ±0.1 |
| FEDDYN | 6.0 ±0.5 | 60.8 ±0.7 | ✗ | 51.9 ±0.2 | 50.7 ±0.2 | 50.8 ±0.2 |
| ADABEST | 8.4 ±2.0 | 55.6 ±0.3 | 35.6 ±0.3 | 49.7 ±0.4 | 47.3 ±0.1 | 47.1 ±0.2 |
| MIME | 26.8 ±2.1 | 59.0 ±0.3 | 45.3 ±0.4 | 50.9 ±0.4 | 48.3 ±0.2 | 48.5 ±0.1 |
| FEDAVGM | 24.8 ±0.7 | 58.7 ±0.9 | 42.1 ±0.3 | 50.7 ±0.2 | 50.0 ±0.0 | 50.4 ±0.1 |
| FEDACG | 25.7 ±0.5 | 58.7 ±0.3 | 43.5 ±0.4 | 51.3 ±0.3 | 50.9 ±0.1 | 51.0 ±0.1 |
| SCAFFOLD-M | 30.9 ±0.7 | 60.1 ±0.5 | 45.7 ±0.2 | 50.1 ±0.3 | 50.8 ±0.0 | 51.0 ±0.1 |
| FEDCM (GHBM $\tau=1$) | 22.2 ±1.0 | 53.1 ±0.2 | 36.0 ±0.3 | 50.2 ±0.5 | 49.2 ±0.1 | 50.4 ±0.1 |
| MIMEMOM | 24.3 ±0.9 | 60.5 ±0.6 | 48.2 ±0.7 | 50.6 ±0.1 | 48.5 ±0.2 | 48.9 ±0.2 |
| MIMELITEMOM | 21.2 ±1.6 | 59.2 ±0.5 | 46.0 ±0.3 | 50.7 ±0.1 | 49.1 ±0.4 | 49.4 ±0.3 |
| LOCALGHBM (ours) | 38.2 ±1.0 | 62.0 ±0.5 | 50.3 ±0.5 | 51.9 ±0.4 | 51.2 ±0.1 | 51.1 ±0.3 |
| FEDHBM (ours) | **42.5** ±0.8 | **62.5** ±0.5 | **50.4** ±0.5 | **52.0** ±0.4 | **51.3** ±0.1 | **51.4** ±0.2 |

Table 2: **Test accuracy (%) comparison of best SOTA FL algorithms on large-scale and realistic settings.** GHBM is the best algorithm when client participation is extremely low, while FEDHBM still improves the other competitors by a large margin. ✗ means that the algorithm did not converge.

| METHOD | MOBILENETV2 | | | | VIT-B\16 | | | |
|---|---|---|---|---|---|---|---|---|
| | GLDv2 | INATURALIST | | | GLDv2 | INATURALIST | | STACKOVERFLOW |
| | $C \approx 0.79\%$ | $C \approx 0.1\%$ | $C \approx 0.5\%$ | $C \approx 1\%$ | $C \approx 0.79\%$ | $C \approx 0.1\%$ | $C \approx 0.5\%$ | $C \approx 0.12\%$ |
| FEDAVG | 60.3 ±0.2 | 38.0 ±0.8 | 45.25 ±0.1 | 47.59 ±0.1 | 68.5 ±0.5 | 65.6 ±0.1 | 70.7 ±0.8 | 24.0 ±0.4 |
| SCAFFOLD | 61.0 ±0.1 | ✗ | ✗ | ✗ | 67.5 ±3.3 | ✗ | ✗ | 24.8 ±0.4 |
| FEDAVGM | 61.5 ±0.2 | 41.3 ±0.4 | 46.0 ±0.1 | 48.4 ±0.1 | 70.0 ±0.5 | 66.0 ±0.2 | 71.4 ±0.5 | 24.1 ±0.3 |
| MIMEMOM | ✗ | ✗ | ✗ | ✗ | ✗ | ✗ | ✗ | 24.9 ±0.6 |
| GHBM - best $\tau$ (ours) | **65.9** ±0.1 | **41.8** ±0.1 | **48.7** ±0.1 | **50.5** ±0.1 | **74.3** ±0.6 | **68.8** ±0.3 | **73.5** ±0.4 | **27.0** ±0.1 |
| FEDHBM (ours) | 65.4 ±0.2 | 41.6 ±0.2 | 47.3 ±0.0 | 49.8 ±0.0 | 73.1 ±0.9 | 66.7 ±0.7 | 72.1 ±0.5 | 24.5 ±0.4 |

Table 3: **Total communication and computational cost for reaching the final model quality of FEDAVG**, across academic and real-world large-scale datasets (details in Appendix C.3). The coloured arrows indicate respectively a reduction (↓) and an increase (↑) of comm./comp. cost.

| METHOD | COMM. OVERHEAD | TOTAL COMMUNICATION COST (BYTES EXCHANGED) | | | | TOTAL COMPUTATIONAL COST (WALL-CLOCK TIME HH:MM) | | | |
|---|---|---|---|---|---|---|---|---|---|
| | | CIFAR-100 ($\alpha = 0$) | | GLDv2 | | CIFAR-100 ($\alpha = 0$) | | GLDv2 | |
| | | CNN | RESNET-20 | MOBILENETV2 | VIT-B\16 | CNN | RESNET-20 | MOBILENETV2 | VIT-B\16 |
| FEDAVG | 1× | 30.9 GB | 10.3 GB | 89.8 GB | 483.7 GB | 02:05 | 03:36 | 13:51 | 13:56 |
| SCAFFOLD | 2× | 40.8 GB ↑32.0% | 14.2 GB ↑37.8% | 51.2 GB ↓43.0% | 967.4 GB ↑100.0% | 01:23 ↓34.0% | 02:39 ↓26.4% | 08:28 ↓38.9% | 15:15 ↑9.4% |
| FEDAVGM | 1× | 21.0 GB ↓32.0% | 9.1 GB ↓11.6% | 73.6 GB ↓18.0% | 403.1 GB ↓16.7% | 01:25 ↓32.0% | 03:10 ↓12.0% | 11:37 ↓16.7% | 11:37 ↓16.7% |
| MIMEMOM | 3× | 21.5 GB ↓30.4% | 30.9 GB ↑200.0% | 269.4 GB ↑200.0% | 1.417 TB ↑200.0% | 01:27 ↓30.4% | 10:42 ↑197.8% | 41:07 ↑197.8% | 41:30 ↑197.8% |
| GHBM (ours) | 1.5× | 8.5 GB ↓72.5% | 7.0 GB ↓32.5% | 48.5 GB ↓46.0% | 314.4 GB ↓35.0% | 00:24 ↓80.8% | 01:37 ↓55.0% | 05:20 ↓61.5% | 06:30 ↓53.3% |
| FEDHBM (ours) | 1× | 5.2 GB ↓83.0% | 4.2 GB ↓59.2% | 29.6 GB ↓67.0% | 234.4 GB ↓51.5% | 00:22 ↓82.0% | 01:29 ↓59.0% | 06:23 ↓54.0% | 07:31 ↓46.0% |

## 6 Conclusions

In this work, we propose *Generalized Heavy-Ball Momentum* (GHBM), a novel momentum-based optimization method for FL that effectively mitigates the joint effect of statistical heterogeneity and partial participation. We theoretically prove that GHBM converges under arbitrary heterogeneity in *cyclic partial participation*, achieving the same rate classical momentum enjoys in *full participation*. Extensive experiments, confirm that GHBM significantly outperforms state-of-the-art FL methods in both convergence speed and final model quality, demonstrating its robustness in large-scale, real-world heterogeneous FL scenarios.

## Funding

Riccardo Zaccone and Carlo Masone declare that financial support was received for the research, authorship, and/or publication of this article. This study was carried out within the project FAIR - Future Artificial Intelligence Research - and received funding from the European Union Next-GenerationEU [PIANO NAZIONALE DI RIPRESA E RESILIENZA (PNRR) – MISSIONE 4 COMPONENTE 2, INVESTIMENTO 1.3 – D.D. 1555 11/10/2022, PE00000013 - CUP: E13C22001800001]. This manuscript reflects only the authors' views and opinions, neither the European Union nor the European Commission can be considered responsible for them. A part of the computational resources for this work was provided by hpc@polito, which is a Project of Academic Computing within the Department of Control and Computer Engineering at the Politecnico di Torino (`http://www.hpc.polito.it`). We acknowledge the CINECA award under the ISCRA initiative for the availability of high-performance computing resources. This work was supported by CINI.

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

# A Additional Discussion

## A.1 Extended Related Works

Recently, similarly based on variance reduction as SCAFFOLD, [Mishchenko et al., 2022] propose SCAFFNEW to achieve accelerated communication complexity in heterogeneous settings through control variates, guaranteeing convergence under arbitrary heterogeneity in full participation. The work by Mishchenko et al. [2024], under the assumption of second-order data heterogeneity, proposes an algorithm which can reduce client drift by estimating the global update direction as well as employing regularization. The proposed algorithm can be seen as a combination of FEDPROX with SCAFFOLD/SCAFFNEW, and similarly relies on additional server control variates to correct the drift, so the underlying principle is still variance reduction. Quite differently, GHBM is based on momentum, properly modified to tackle heterogeneity and partial participation in FL. Similarly to the already discussed MIME [Karimireddy et al., 2021], Karagulyan et al. [2024] propose the SPAM algorithm and leverage momentum as a local correction term to benefit from second-order similarity.

**Lowering Communication Requirements in FL.** Researchers have studied methods to reduce the memory needed for exchanging gradients in the distributed setting, for example by quantization [Alistarh et al., 2017] or by compression [Mishchenko et al., 2019, Koloskova et al., 2020]. In the context of FL, such ideas have been developed to meet the communication and scalability constraints [Reisizadeh et al., 2020], and to take into account heterogeneity [Sattler et al., 2020]. With a similar idea, quantization has been incorporated into a recent momentum-based FL approach [Das et al., 2022] to limit the communication overhead, still requiring significantly more computation client-side. Our work focuses on a novel formulation of momentum that takes into account the joint effects of heterogeneity and partial participation, and that has a heavy-ball structure allowing efficient use of the information already being sent in vanilla FEDAVG, so additional techniques to compress that information remain orthogonal to our approach.

**Comparison with FedACG [Kim et al., 2024].** We provide a comparison with the FedACG algorithm based on: algorithmic design, theoretical guarantees and empirical results. Algorithmically, it has two modifications w.r.t. FEDAVGM: (i) it uses the Nesterov Accelerated Gradient (NAG) to broadcast a lookahead global model and (ii) adds a proximal local penalty similar to FEDPROX w.r.t. this transmitted global model. The method has the same communication complexity as FedAvg, because it does not exchange additional information. Our work proposes instead a novel formulation of momentum, explicitly designed to provide an advantage in heterogeneous FL with partial client participation. We propose both the main algorithm (GHBM), which has *stateless* clients but has $1.5\times$ the communication complexity of FedAvg, and communication efficient versions (*e.g.* FEDHBM), that preserve the communication complexity as FedAvg, at the cost of using local storage. From a theoretical perspective, the convergence rate of FedACG does not prove any advantage w.r.t. heterogeneity, since it still relies on the bounded heterogeneity assumption. GHBM is proven to converge under arbitrary heterogeneity in cyclic partial participation, recovering the same convergence rate that Cheng et al. [2024] proved for FEDCM when in full participation. This is a significant advantage that then reflects in significantly improved performance. From an empirical perspective, simulation results are presented in Fig. 7. While it is faster than FedAvgM, it still falls short behind our algorithms in heterogeneous scenarios. This is a consequence of the same issue we showed in Sec. 3.2 for classical momentum.

## A.2 Advantage of Local Steps and Connections to Incremental Gradient Methods.

Thm. 4.6 does not show an explicit benefit from the local steps, similar to the best-known theory for momentum-based FL methods [Cheng et al., 2024]. However, GHBM offers a clear advantage w.r.t. centralized methods for finite-sum optimization applied in FL (where clients represent functions), referred to as *incremental gradient methods*. One algorithm of this family, the Incremental Aggregated Gradient (IAG), removes the effect of functions heterogeneity by approximating a full gradient with an aggregate of past gradients, assuming cyclic participation [Gürbüzbalaban et al., 2015]. However, this holds only in standard distributed mini-batch optimization, where $J = 1$. GHBM shares a similar intuition, but applying this logic to the momentum update rather than the gradient estimate is crucial when local steps are involved. Simply extending IAG with local steps would not mitigate client drift-induced heterogeneity as GHBM does. In fact, its convergence rate would be bounded by that of FEDAVG in full participation, whose lower bound is known to be affected by heterogeneity (see Thm. II of Karimireddy et al. [2020]).

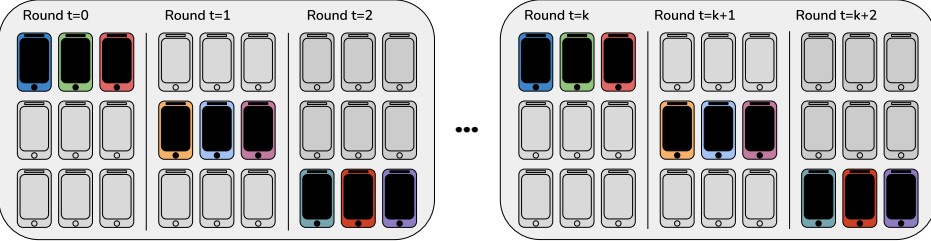

Cyclic participation with period p=3 for any round k s.t. k mod p = 0

Figure 4: **Illustration of cyclic client participation with a total of** $K = 9$ **clients.** Thm. 4.6 holds under the assumption of cyclic participation, which simply states that there is any fixed order (so client shuffling methods like Shuffle-Once are compliant with the assumption) in which clients appear across rounds in the training, *i.e.* each client is sampled every $p = \frac{1}{C}$ rounds. In the above image, $K \cdot C = 3$ clients are selected for training, *i.e.* each client is selected exactly once every $p = 3$ rounds.

## A.3 On the Use of Cyclic Participation Assumption.

---

**Algorithm 1:** GHBM, LOCALGHBM and FEDAVG

---

**Require:** initial model $\theta^0$, $K$ clients, $C$ participation ratio, $T$ number of total round, $\eta$ and $\eta_l$ learning rates, $\tau \in \mathbb{N}^+$.

1: **for** $t = 1$ to $T$ **do**
2:    $\mathcal{S}^t \leftarrow$ subset of clients $\sim \mathcal{U}(\mathcal{S}, \max(1, K \cdot C))$
3:    Send $\theta^{t-1}, \theta^{t-\tau-1}$ to all clients $i \in \mathcal{S}^t$
4:    **for** $i \in \mathcal{S}^t$ **in parallel do**
5:       $\theta_i^{t,0} \leftarrow \theta^{t-1}$
6:       Retrieve $\theta^{t-\tau_i-1}$ from local storage
7:       $\tilde{m}_\tau^t \leftarrow \frac{1}{\tau J}(\theta^{t-1} - \theta^{t-\tau-1})$
8:       $\tilde{m}_{\tau_i}^t \leftarrow \frac{1}{\tau_i J}(\theta^{t-1} - \theta^{t-\tau_i-1})$ **if** $\theta^{t-\tau_i-1}$ is set **else 0**
9:       **for** $j = 1$ to $J$ **do**
10:          sample a mini-batch $d_{i,j}$ from $\mathcal{D}_i$
11:          $\theta_i^{t,j} \leftarrow \theta_i^{t,j-1} - \eta_l \tilde{g}_i^{t,j} + \beta \tilde{m}_\tau^t + \beta \tilde{m}_{\tau_i}^t$
12:       **end for**
13:       Save model $\theta^{t-1}$ into local storage
14:    **end for**
15:    $\tilde{g}^t \leftarrow \frac{1}{|\mathcal{S}^t|} \sum_{i \in \mathcal{S}^t} \left( \theta^{t-1} - \theta_i^{t,J} \right)$
16:    $\theta^t \leftarrow \theta^{t-1} - \eta \tilde{g}^t$
17: **end for**

---

The use of cyclic participation in the proof of Thm. 4.6 allows precise control over the clients' contributions to the average of the last $\tau$ pseudo-gradients. This ensures that the $\tau$-averaged pseudo-gradient used to update the momentum is unaffected by heterogeneity, which is the important point behind the proof of Thm. 4.6. Under random uniform, due to the non-zero probability of sampling the same client within $\tau$ rounds, this condition is hardly verified. Although one could technically enforce this condition without cyclic sampling — by explicitly tracking each client's pseudo-gradient and computing a uniform average across the most recent one from each client — this would be impractical. Such a design would not be compliant with protocols like Secure Aggregation, widely adopted in real-world FL systems, thus posing a significant practical limitation. Please note that in our analysis convergence under unbounded heterogeneity is not a simple byproduct of the assumption, but comes explicitly from the algorithmic structure of GHBM (*i.e.* setting $\tau = \frac{k}{C}, \forall k \in \mathbb{N}^+$ is **necessary**). The best-known analysis of FEDAVG under cyclic participation is provided by Cho et al. [2023], which proves that in certain situations (*e.g.* clients run GD instead of SGD) there can be an asymptotic advantage in the case we prospect with Assumption 4.4. However, it is important to notice that all the results presented in Cho et al. [2023] rely on forms of bounded heterogeneity, and with this respect, the results presented in this work are novel and advance state of the art.

## A.4 Applicability of GHBM-based Algorithms in FL Scenarios.

Although based on the same principle, our algorithms are suitable for different scenarios. Similarly to algorithms proposed for cross-device FL [Karimireddy et al., 2021], GHBM uses *stateless* clients, with the main $\tau$ hyperparameter controlled by the server. This ensures that clients always apply a momentum term consistent with the GHBM update rule, differently from algorithms that require clients participating in multiple rounds to adhere to their formulation, such as SCAFFOLD and FEDDYN. This is particularly important when the number of clients is large and a small portion of them participates in each round, and it is why, in our large-scale setting, these methods fail to converge.

Table 4: **Comparison of convergence rates of FL algorithms.** GHBM improves the state-of-art by attaining, in *cyclic partial participation*, the same rate of classical momentum in *full participation*. Remind that $L$ is the smoothness constant of objective functions, $\Delta = f(\theta^0) - \min_\theta f(\theta)$ is the initialization gap, $\sigma^2$ is the clients' gradient variance, $|\mathcal{S}|$ is the number of clients, $C$ is the participation ratio, $J$ is the number of local steps per round, and $T$ is the number of communication rounds. $\zeta := \sup_\theta \|\nabla f(\theta)\|$ and $G$ are uniform bounds of gradient norm and dissimilarity.

| Algorithm | Convergence Rate $\frac{1}{T}\sum_{t=1}^{T}\mathbb{E}\left[\|\nabla f(\theta^t)\|^2\right] \lesssim$ | Additional Assumptions | Partial participation? |
|---|---|---|---|
| FEDAVG [Yang et al., 2021] | $\left(\frac{L\Delta\sigma^2}{\|\mathcal{S}\|JT}\right)^{1/2} + \frac{L\Delta}{T}$ | Bounded hetero.[1] | ✗ |
| [Yang et al., 2021] | $\left(\frac{L\Delta J\sigma^2}{\|\mathcal{S}\|CT}\right)^{1/2} + \frac{L\Delta}{T}$ | Bounded hetero.[1] | ✓ |
| FEDCM [Xu et al., 2021] | $\left(\frac{L\Delta(\sigma^2+\|\mathcal{S}\|CJ\zeta^2)}{\|\mathcal{S}\|CJT}\right)^{1/2} + \left(\frac{L\Delta(\sigma/\sqrt{J}+\sqrt{\|\mathcal{S}\|C}(\zeta+G))}{\sqrt{\|\mathcal{S}\|C}T}\right)^{2/3}$ | Bounded grad. Bounded hetero. | ✓ |
| [Cheng et al., 2024] | $\left(\frac{L\Delta\sigma^2}{\|\mathcal{S}\|JT}\right)^{1/2} + \frac{L\Delta}{T}$ | – | ✗ |
| SCAFFOLD-M [Cheng et al., 2024] | $\left(\frac{L\Delta\sigma^2}{\|\mathcal{S}\|CJT}\right)^{1/2} + \frac{L\Delta}{T}\left(1+\frac{\|\mathcal{S}\|^{2/3}}{\|\mathcal{S}\|C}\right)$ | – | ✓ |
| **GHBM (Thm. 4.6)** | $\left(\frac{L\Delta\sigma^2}{\|\mathcal{S}\|JT}\right)^{1/2} + \frac{L\Delta}{T}$ | Cyclic participation | ✓ |

[1] The local learning rate vanishes to zero when gradient dissimilarity is unbounded, *i.e.*, $G \to \infty$.

These design choices make our algorithm in practice suitable for cross-device FL, where it offers significant advantages, as experimentally validated in Sec. 5.2. On the other hand, FEDHBM and LOCALGHBM take advantage of the fact that clients participate multiple times in the training process to remove the need to send the momentum term from the server, recovering the same communication complexity of FEDAVG. As a result, clients in these methods are *stateful* - requiring to maintain variables across rounds [Kairouz et al., 2021] - and are therefore best suited for scenarios akin to *cross-silo* FL.

## A.5 Theoretical Comparison with other FL algorithms

**Comparison with SCAFFOLD-M.** Recently Cheng et al. [2024] proved that momentum accelerates SCAFFOLD, preserving strong guarantees against heterogeneity in partial participation. However, the resulting SCAFFOLD-M method is still based on variance reduction, *i.e.*, it converges under arbitrary heterogeneity thanks to variance reduction, not because it uses momentum. Our rate additionally requires Assumption 4.4, but is faster and, most importantly, shows that momentum, when modified according to our formulation, can by itself provide similar guarantees even when not all clients participate.

## A.6 Notes on Failure Cases of SOTA Algorithms

In this paper, we evaluated our approach using the large-scale FL datasets proposed by [Hsu et al., 2020]. Notably, several recent state-of-the-art FL algorithms failed to converge on these datasets. For SCAFFOLD this result aligns with prior works [Reddi et al., 2021, Karimireddy et al., 2021], since it is unsuitable for cross-device FL with thousands of devices. Indeed, the client control variates can become stale, and may consequently degrade the performance. For MIMEMOM [Karimireddy et al., 2021], despite extensive hyperparameter tuning using the authors' original code, we were unable to achieve convergence. This finding is surprising since the approach has been proposed to tackle cross-device FL. To our knowledge, this is the first work to report these failure cases, likely due to the lack of prior evaluations on such challenging datasets. We believe these findings underscore the need for further investigation into the factors contributing to algorithm performance in large-scale, heterogeneous FL settings.

## B   Proofs

**Algorithms**

To handle the proof, we analyze a simpler version of our algorithm, in which we use the update rule in Eq. (5) instead of the one described in Eq. (6). The resulting Algorithm 3 we analyze is reported along the plain GHBM (Algorithm 2) we used in the experiments. Both algorithms enjoy the same underlying idea: use the gradients of a larger portion of the clients to estimate the momentum term.

---

**Algorithm 2:** GHBM (PRACTICAL VERSION)

---

**Require:** initial model $\theta^0$, $K$ clients, $C$ participation ratio, $T$ number of total round, $\eta$ and $\eta_l$ learning rates, $\tau \in \mathbb{N}^+$.

1: **for** $t = 1$ to $T$ **do**
2:   $\mathcal{S}^t \leftarrow$ subset of clients $\sim \mathcal{U}(\mathcal{S}, \max(1, K \cdot C))$
3:   **for** $i \in \mathcal{S}^t$ **in parallel do**
4:     $\theta_i^{t,0} \leftarrow \theta^{t-1}$
5:     **for** $j = 1$ to $J$ **do**
6:       sample a mini-batch $d_{i,j}$ from $\mathcal{D}_i$
7:       $u_i^{t,j} \leftarrow \nabla f_i(\theta_i^{t,j-1}, d_{i,j}) + \beta \tilde{m}_\tau^t$
8:       $\theta_i^{t,j} \leftarrow \theta_i^{t,j-1} - \eta_l u_i^{t,j}$
9:     **end for**
10:   **end for**
11:   $u^t \leftarrow \frac{1}{|\mathcal{S}^t|} \sum_{i \in \mathcal{S}^t} \left( \theta^{t-1} - \theta_i^{t,J} \right)$
12:   $\theta^t \leftarrow \theta^{t-1} - \eta u^t$
13:   $\tilde{m}_\tau^{t+1} \leftarrow \frac{1}{\tau J} \left( \theta^{t-\tau} - \theta^t \right)$
14: **end for**

---

**Algorithm 3:** GHBM (THEORY VERSION)

---

**Require:** initial model $\theta^0$, $K$ clients, $C$ participation ratio, $T$ number of total round, $\eta$ and $\eta_l$ learning rates, $\tau \in \mathbb{N}^+$.

1: **for** $t = 1$ to $T$ **do**
2:   $\mathcal{S}^t \leftarrow$ subset of clients $\sim \mathcal{U}(\mathcal{S}, \max(1, K \cdot C))$
3:   **for** $i \in \mathcal{S}^t$ **in parallel do**
4:     $\theta_i^{t,0} \leftarrow \theta^{t-1}$
5:     **for** $j = 1$ to $J$ **do**
6:       sample a mini-batch $d_{i,j}$ from $\mathcal{D}_i$
7:       $u_i^{t,j} \leftarrow \beta \nabla f_i(\theta_i^{t,j-1}, d_{i,j}) + (1 - \beta)\tilde{m}_\tau^t$
8:       $\theta_i^{t,j} \leftarrow \theta_i^{t,j-1} - \eta_l u_i^{t,j}$
9:     **end for**
10:   **end for**
11:   $u^t \leftarrow \frac{1}{\eta_l |\mathcal{S}^t| J} \sum_{i \in \mathcal{S}^t} \left( \theta^{t-1} - \theta_i^{t,J} \right)$
12:   $\bar{\theta}^t \leftarrow \theta^{t-1} - u^t + (1 - \beta)\tilde{m}_\tau^t$
13:   $\tilde{m}_\tau^{t+1} \leftarrow (1 - \beta)\tilde{m}_\tau^t + \frac{1}{\tau} \left( \bar{\theta}^{t-\tau} - \bar{\theta}^t \right)$
14:   $\theta^t \leftarrow \theta^{t-1} - \eta \tilde{m}_\tau^{t+1}$
15: **end for**

---

In the following, we list the differences between the two:

1. Explicit use of $\tau$-averaged gradients when updating the momentum term (line 13). This can be implemented by keeping server-side an auxiliary sequence of models $\bar{\theta}^t$, in which the momentum added client side is subtracted server-side (line 12), such that taking the difference of two models gives the sum of pseudo-grads.

2. Use of convex sum in local updates (line 7). This is done to align with the formulation of momentum methods in Cheng et al. [2024], and more in general with the formulation of momentum commonly analyzed in literature. There is no theoretical difference between the two versions, as they only differ by a constant scaling [Liu et al., 2020].

3. Use of gradients averaged over local steps (line 11). This is done to align with the analysis of Cheng et al. [2024], Xu et al. [2021], and it is equivalent to coupling server and client learning rates (*i.e.* setting $\eta = \gamma J \eta_l$ in Algorithm 3, where $\gamma$ is the server learning rate we would use in Algorithm 2).

The two algorithms have similar performances, which are reported in Fig. 5

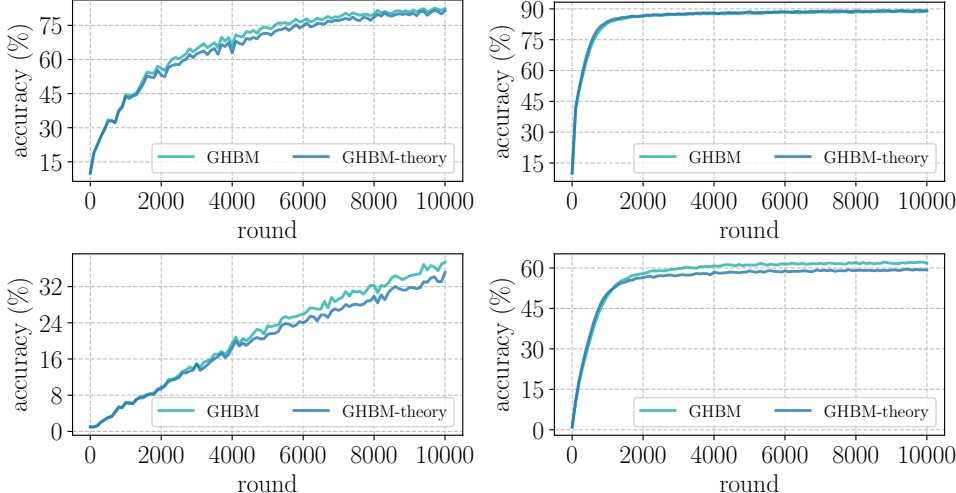

Figure 5: Comparing the GHBM implementation analyzed in theory (Algorithm 3) with the one proposed in the main paper (Algorithm 2). The plots show the convergence rate on CIFAR-10 (top) and CIFAR-100 (bottom), in NON-IID (left) and IID (right) scenarios with RESNET-20 architecture.

### Preliminaries

Our convergence proof for GHBM is based on the recent work of Cheng et al. [2024], which offers new proof techniques for momentum-based FL algorithms. Throughout the proofs we use the following auxiliary variables to facilitate the presentation:

$$\mathcal{U}_t := \frac{1}{|\mathcal{S}|J} \sum_{j=1}^{J} \sum_{i=1}^{|\mathcal{S}|} \mathbb{E}\left[\left\|\theta_i^{t,j} - \theta^{t-1}\right\|^2\right] \tag{9}$$

$$\mathcal{E}_t := \mathbb{E}\left[\left\|\nabla f(\theta^{t-1}) - \tilde{m}_\tau^{t+1}\right\|^2\right] \tag{10}$$

$$\zeta_i^{t,j} := \mathbb{E}\left[\theta_i^{t,j+1} - \theta_i^{t,j}\right] \tag{11}$$

$$\Xi_t := \frac{1}{|\mathcal{S}|} \sum_{i=1}^{|\mathcal{S}|} \mathbb{E}\left[\left\|\zeta_i^{t,0}\right\|^2\right]$$

$$\Lambda_t := \mathbb{E}\left[\left\|\left(\frac{1}{\tau} \sum_{k=t-\tau+1}^{t} \frac{1}{|\mathcal{S}^k|J} \sum_{i=1}^{|\mathcal{S}^k|} \sum_{j=1}^{J} \tilde{g}_i^{k,j}(\theta_i^{k,j-1})\right) - g^{t_\tau}\right\|^2\right] \tag{12}$$

$$\gamma_t := \mathbb{E}\left[\left\|g^{t_\tau} - \nabla f(\theta^{t-1})\right\|^2\right] \tag{13}$$

Additionally, here we report the *bounded gradient heterogeneity* assumption. It is used to quantify the heterogeneity reduction effect of GHBM varying its $\tau$ hyperparameter. Notice that our main claim does not depend on this assumption, as for the optimal value of $\tau = 1/C$ the assumption is not needed (see Lemma B.4).

### B.1 Momentum Expressions

In this section we report the derivation of the momentum expressions in Eq. (3) and (7) from the main paper.

**Lemma B.1** (Heavy-Ball Formulation of Classical Momentum). *Let us consider the following classical formulation of momentum:*

$$\tilde{m}^t = \beta \tilde{m}^{t-1} + \tilde{g}^t(\theta^{t-1}) \tag{14}$$

$$\theta^t = \theta^{t-1} - \eta \tilde{m}^t \tag{15}$$

*The same update rule can be equivalently expressed with the following, known as heavy-ball formulation:*

$$\theta^t = \theta^{t-1} + \beta(\theta^{t-1} - \theta^{t-2}) - \eta \tilde{g}(\theta^{t-1}) \tag{16}$$

*Proof.* First derive the expression of $\tilde{m}^t$ from Eq. (15), both for time $t$ and $t-1$:

$$\tilde{m}^t = \frac{(\theta^{t-1} - \theta^t)}{\eta}$$

$$\tilde{m}^{t-1} = \frac{(\theta^{t-2} - \theta^{t-1})}{\eta}$$

Now plug these expressions into Eq. (14) to obtain (16):

$$\frac{(\theta^{t-1} - \theta^t)}{\eta} = \beta \frac{(\theta^{t-2} - \theta^{t-1})}{\eta} + \tilde{g}^t(\theta^{t-1})$$

$$(\theta^t - \theta^{t-1}) = \beta (\theta^{t-1} - \theta^{t-2}) - \eta \tilde{g}^t(\theta^{t-1})$$

$$\theta^t = \theta^{t-1} + \beta (\theta^{t-1} - \theta^{t-2}) - \eta \tilde{g}^t(\theta^{t-1})$$

$\square$

**Lemma B.2** (Heavy-Ball formulation of generalized momentum). *Let us consider the following generalized formulation of momentum:*

$$\tilde{m}^t_\tau = \frac{1}{\tau} \sum_{k=1}^{\tau} \beta \tilde{m}^{t-k}_\tau + \tilde{g}^t(\theta^{t-1}) \tag{17}$$

$$\theta^t = \theta^{t-1} - \eta \tilde{m}^t_\tau \tag{18}$$

*The same update rule can be equivalently expressed in an heavy ball form, which we call as Generalized Heavy-Ball momentum (GHBM):*

$$\theta^t = \theta^{t-1} + \frac{\beta}{\tau}(\theta^{t-1} - \theta^{t-\tau-1}) - \eta \tilde{g}(\theta^{t-1}) \tag{19}$$

*Proof.* First derive the expression of $\tilde{m}^t_\tau$ from Eq. (18), both for time $t$ and $t-1$:

$$\tilde{m}^t_\tau = \frac{(\theta^{t-1} - \theta^t)}{\eta}$$

$$\tilde{m}^{t-1}_\tau = \frac{(\theta^{t-2} - \theta^{t-1})}{\eta}$$

Now plug these expressions into Eq. (17):

$$\frac{(\theta^{t-1} - \theta^t)}{\eta} = \frac{\beta}{\tau} \sum_{k=1}^{\tau} \frac{(\theta^{t-k-1} - \theta^{t-k})}{\eta} + \tilde{g}^t(\theta^{t-1})$$

$$(\theta^t - \theta^{t-1}) = \frac{\beta}{\tau} \sum_{k=1}^{\tau} (\theta^{t-k} - \theta^{t-k-1}) - \eta \tilde{g}^t(\theta^{t-1})$$

$$\theta^t = \theta^{t-1} + \frac{\beta}{\tau} \sum_{k=1}^{\tau} (\theta^{t-k} - \theta^{t-k-1}) - \eta \tilde{g}^t(\theta^{t-1})$$

$$\theta^t = \theta^{t-1} + \frac{\beta}{\tau}(\theta^{t-1} - \theta^{t-\tau-1}) - \eta \tilde{g}^t(\theta^{t-1})$$

Where the last equality (19) comes from telescoping the summation on the rhs. $\square$

## B.2 Technical Lemmas

Now we cover some technical lemmas which are useful for computations later on. These are known results that are reported here for the convenience of the reader.

**Lemma B.3** (relaxed triangle inequality). *Let $\{\boldsymbol{v}_1, \dots, \boldsymbol{v}_n\}$ be $n$ vectors in $\mathbb{R}^d$. Then, the following is true:*

$$\left\| \sum_{i=1}^{n} \boldsymbol{v}_i \right\|^2 \leq n \sum_{i=1}^{n} \|\boldsymbol{v}_i\|^2$$

*Proof.* By Jensen's inequality, given a convex function $\phi$, a series of $n$ vectors $\{\boldsymbol{v}_1, \dots, \boldsymbol{v}_n\}$ and a series of non-negative coefficients $\lambda_i$ with $\sum_{i=1}^{n} \lambda_i = 1$, it results that

$$\phi \left( \sum_{i=1}^{n} \lambda_i \boldsymbol{v}_i \right) \leq \sum_{i=1}^{n} \lambda_i \phi \left( \boldsymbol{v}_i \right)$$

Since the function $\boldsymbol{v} \to \|\boldsymbol{v}\|^2$ is convex, we can use this inequality with coefficients $\lambda_1 = \dots = \lambda_n = 1/n$, with $\sum_{i=1}^{n} \lambda_i = 1$, and obtain that

$$\left\| \frac{1}{n} \sum_{i=1}^{n} \boldsymbol{v}_i \right\|^2 = \frac{1}{n^2} \left\| \sum_{i=1}^{n} \boldsymbol{v}_i \right\|^2 \leq \frac{1}{n} \sum_{i=1}^{n} \|\boldsymbol{v}_i\|^2$$

$\square$

## B.3 Proofs of Main Lemmas

In this section we provide the proofs of the main theoretical results presented in the main paper.

**Lemma B.4** (Deviation of $\tau$-averaged gradient from true gradient). *Define $\mathcal{S}_\tau^t := \cup_{k=0}^{\tau-1} \mathcal{S}^{t-k}$ as the set of clients selected in the last $\tau$ rounds, and $g^{t_\tau} := 1/|\mathcal{S}_\tau^t| \sum_{i=1}^{|\mathcal{S}_\tau^t|} g_i^t(\theta^{t-1})$ as the average server pseudo-gradient. The approximation of a gradient over the last $\tau$ rounds $g^{t_\tau}$ w.r.t. the true gradient is quantified by the following:*

$$\mathbb{E} \left[ \|g^{t_\tau} - \nabla f(\theta^{t-1})\|^2 \right] \leq 8 \mathbb{E} \left[ \left( \frac{|\mathcal{S}| - |\mathcal{S}_\tau^t|}{|\mathcal{S}|} \right)^2 \right] \left( G^2 + \|\nabla f(\theta^{t-1})\|^2 \right)$$

**Proof of Lemma B.4** (Deviation of $\tau$-averaged gradient from true gradient)

Let define $\mathcal{S}_d := \mathcal{S} - \mathcal{S}_\tau^t$ and $\mathcal{S}_i := \mathcal{S} \cap \mathcal{S}_\tau^t$. Let us note that when all clients participate, *i.e.* $\mathcal{S}_d = \emptyset$, the claim is trivially true. For $\mathcal{S}_d \neq \emptyset$, we can expand the terms at the left-hand side using their definitions as follows:

$$\gamma_t = \mathbb{E} \left[ \left\| \frac{1}{|\mathcal{S}_\tau^t|} \sum_{i=1}^{|\mathcal{S}_\tau^t|} g_i^t - \frac{1}{|\mathcal{S}|} \sum_{i=1}^{|\mathcal{S}|} g_i^t \right\|^2 \right] \tag{20}$$

$$= \mathbb{E} \left[ \left\| \sum_{i \in \mathcal{S}_i} \left( \frac{1}{|\mathcal{S}_\tau^t|} - \frac{1}{|\mathcal{S}|} \right) g_i^t - \sum_{k \in \mathcal{S}_d} \frac{1}{|\mathcal{S}|} g_k^t \right\|^2 \right] \tag{21}$$

$$\overset{\text{lemma B.3}}{\leq} 2 \left( \underbrace{\mathbb{E} \left[ \left\| \sum_{i \in \mathcal{S}_i} \left( \frac{1}{|\mathcal{S}_\tau^t|} - \frac{1}{|\mathcal{S}|} \right) g_i^t \right\|^2 \right]}_{\mathcal{T}_3} + \underbrace{\mathbb{E} \left[ \left\| \sum_{k \in \mathcal{S}_d} \frac{1}{|\mathcal{S}|} g_k^t \right\|^2 \right]}_{\mathcal{T}_4} \right) \tag{22}$$

Let us consider first $\mathcal{T}_3$. We have:

$$\mathcal{T}_3 = \mathbb{E}\left[\left\|\sum_{i \in \mathcal{S}_i}\left(\frac{1}{|\mathcal{S}_\tau^t|} - \frac{1}{|\mathcal{S}|}\right)g_i^t\right\|^2\right] = \mathbb{E}\left[\left(\frac{1}{|\mathcal{S}_\tau^t|} - \frac{1}{|\mathcal{S}|}\right)^2\left\|\sum_{i \in \mathcal{S}_i}g_i^t\right\|^2\right] \tag{23}$$

$$\overset{\text{lemma B.3}}{\leq} \mathbb{E}\left[\left(\frac{1}{|\mathcal{S}_\tau^t|} - \frac{1}{|\mathcal{S}|}\right)^2|\mathcal{S}_i|\sum_{i \in \mathcal{S}_i}\left\|g_i^t\right\|^2\right] \tag{24}$$

$$= \mathbb{E}\left[\left(\frac{1}{|\mathcal{S}_\tau^t|} - \frac{1}{|\mathcal{S}|}\right)^2|\mathcal{S}_i|\sum_{i \in \mathcal{S}_i}\left\|g_i^t - \nabla f(\theta^{t-1}) + \nabla f(\theta^{t-1})\right\|^2\right] \tag{25}$$

$$\overset{\text{lemma B.3}}{\leq} 2\mathbb{E}\left[\left(\frac{1}{|\mathcal{S}_\tau^t|} - \frac{1}{|\mathcal{S}|}\right)^2|\mathcal{S}_i|\sum_{i \in \mathcal{S}_i}\left(\left\|g_i^t - \nabla f(\theta^{t-1})\right\|^2 + \left\|\nabla f(\theta^{t-1})\right\|^2\right)\right] \tag{26}$$

$$\overset{\text{assumption 4.3}}{\leq} 2\mathbb{E}\left[\left(\frac{1}{|\mathcal{S}_\tau^t|} - \frac{1}{|\mathcal{S}|}\right)^2|\mathcal{S}_i|\left(|\mathcal{S}_i|G^2 + \sum_{i \in \mathcal{S}_i}\left\|\nabla f(\theta^{t-1})\right\|^2\right)\right] \tag{27}$$

Since the term $\nabla f(\theta^{t-1})$ does not depend on the index $i$, we get

$$2\mathbb{E}\left[\left(\frac{1}{|\mathcal{S}_\tau^t|} - \frac{1}{|\mathcal{S}|}\right)^2|\mathcal{S}_i|\left(|\mathcal{S}_i|G^2 + \sum_{i \in \mathcal{S}_i}\left\|\nabla f(\theta^{t-1})\right\|^2\right)\right] \tag{28}$$

$$= 2\mathbb{E}\left[\left(\frac{1}{|\mathcal{S}_\tau^t|} - \frac{1}{|\mathcal{S}|}\right)^2|\mathcal{S}_i|\left(|\mathcal{S}_i|G^2 + |\mathcal{S}_i|\left\|\nabla f(\theta^{t-1})\right\|^2\right)\right] \tag{29}$$

$$= 2\mathbb{E}\left[\left(\frac{1}{|\mathcal{S}_\tau^t|} - \frac{1}{|\mathcal{S}|}\right)^2|\mathcal{S}_i|^2\right]\left(G^2 + \left\|\nabla f(\theta^{t-1})\right\|^2\right) \tag{30}$$

Now, note that $\mathcal{S}_\tau^t \subseteq \mathcal{S} \implies |\mathcal{S}_i| = |\mathcal{S}_\tau^t|$. Therefore,

$$\mathcal{T}_3 \leq 2\mathbb{E}\left[\left(\frac{1}{|\mathcal{S}_\tau^t|} - \frac{1}{|\mathcal{S}|}\right)^2|\mathcal{S}_i|^2\right]\left(G^2 + \left\|\nabla f(\theta^{t-1})\right\|^2\right) \tag{31}$$

$$= 2\mathbb{E}\left[\left(\frac{|\mathcal{S}| - |\mathcal{S}_\tau^t|}{|\mathcal{S}|}\right)^2\right]\left(G^2 + \left\|\nabla f(\theta^{t-1})\right\|^2\right) \tag{32}$$

Moving now to $\mathcal{T}_4$, we have:

$$\mathcal{T}_4 = \mathbb{E}\left[\left\|\sum_{k\in\mathcal{S}_d}\frac{1}{|\mathcal{S}|}g_k^t\right\|^2\right] \leq \mathbb{E}\left[\left(\frac{1}{|\mathcal{S}|}\right)^2\left\|\sum_{k\in\mathcal{S}_d}g_k^t\right\|^2\right] \tag{33}$$

$$\overset{\text{lemma B.3}}{\leq} \mathbb{E}\left[\left(\frac{1}{|\mathcal{S}|}\right)^2|\mathcal{S}_d|\sum_{k\in\mathcal{S}_d}\left\|g_k^t\right\|^2\right] \tag{34}$$

$$= \mathbb{E}\left[\left(\frac{1}{|\mathcal{S}|}\right)^2|\mathcal{S}_d|\sum_{k\in\mathcal{S}_d}\left\|g_k^t - \nabla f(\theta^{t-1}) + \nabla f(\theta^{t-1})\right\|^2\right] \tag{35}$$

$$\overset{\text{lemma B.3}}{\leq} 2\mathbb{E}\left[\left(\frac{1}{|\mathcal{S}|}\right)^2|\mathcal{S}_d|\sum_{k\in\mathcal{S}_d}\left(\left\|g_k^t - \nabla f(\theta^{t-1})\right\|^2 + \left\|\nabla f(\theta^{t-1})\right\|^2\right)\right] \tag{36}$$

$$\overset{\text{assumption 4.3}}{\leq} 2\mathbb{E}\left[\left(\frac{1}{|\mathcal{S}|}\right)^2|\mathcal{S}_d|\left(|\mathcal{S}_d|G^2 + \sum_{k\in\mathcal{S}_d}\left\|\nabla f(\theta^{t-1})\right\|^2\right)\right] \tag{37}$$

$$= 2\mathbb{E}\left[\left(\frac{1}{|\mathcal{S}|}\right)^2|\mathcal{S}_d|\left(|\mathcal{S}_d|G^2 + |\mathcal{S}_d|\left\|\nabla f(\theta^{t-1})\right\|^2\right)\right] \tag{38}$$

$$= 2\mathbb{E}\left[\left(\frac{|\mathcal{S}_d|}{|\mathcal{S}|}\right)^2\right]\left(G^2 + \left\|\nabla f(\theta^{t-1})\right\|^2\right) \tag{39}$$

$$\tag{40}$$

Observing that $|\mathcal{S}_d| = |\mathcal{S}| - |\mathcal{S}_\tau^t|$ we obtain:

$$\mathcal{T}_4 \leq 2\mathbb{E}\left[\left(\frac{|\mathcal{S}_d|}{|\mathcal{S}|}\right)^2\right]\left(G^2 + \left\|\nabla f(\theta^{t-1})\right\|^2\right) = \mathbb{E}\left[\left(\frac{|\mathcal{S}| - |\mathcal{S}_\tau^t|}{|\mathcal{S}|}\right)^2\right]\left(G^2 + \left\|\nabla f(\theta^{t-1})\right\|^2\right) \tag{41}$$

Finally, by plugging (31) and (41) in (22) we obtain

$$\mathbb{E}_{\mathcal{S}^t\sim\mathcal{U}(\mathcal{S})}\left[\left\|g^{(t)\tau}(\theta) - \nabla f(\theta)\right\|^2\right] \leq 8\mathbb{E}_{\mathcal{S}^t\sim\mathcal{U}(\mathcal{S})}\left[\left(\frac{|\mathcal{S}| - |\mathcal{S}_\tau^t|}{|\mathcal{S}|}\right)^2\right]\left(G^2 + \left\|\nabla f(\theta)\right\|^2\right)$$

which concludes the proof.

$\square$

**Corollary B.5.** *Consider Lemma B.4 and further assume that, at each round of FL training, clients are sampled according to a rule satisfying Assumption 4.4. Then, for any $\tau \in \left(0, \frac{1}{C}\right]$:*

$$\mathbb{E}\left[\left\|g^{t_\tau} - \nabla f(\theta^{t-1})\right\|^2\right] \leq 8\left(1 - \tau C\right)^2\left(G^2 + \left\|\nabla f(\theta^{t-1})\right\|^2\right)$$

**Proof of Corollary B.5**    This corollary follows from Lemma B.4, which states that

$$\mathbb{E}_{\mathcal{S}^t\sim\mathcal{U}(\mathcal{S})}\left[\left\|g^{(t)\tau}(\theta) - \nabla f(\theta)\right\|^2\right] \leq 8\mathbb{E}_{\mathcal{S}^t\sim\mathcal{U}(\mathcal{S})}\left[\left(\frac{|\mathcal{S}| - |\mathcal{S}_\tau^t|}{|\mathcal{S}|}\right)^2\right]\left(G^2 + \left\|\nabla f(\theta)\right\|^2\right)$$

To prove the results, we use (i) Assumption 4.4, (ii) the fact that $|\mathcal{S}^t| = |\mathcal{S}|C \ \forall t$ and (iii) $\mathcal{S}_\tau^t$ is union of $\tau$ disjoint $\mathcal{S}^t$ sets. Using points (i)-(iii), and assuming $\tau \in [0, \frac{1}{C}]$, it follows that:

$$\left\|g^{(t)\tau}(\theta) - \nabla f(\theta)\right\|^2 \leq 8\left(1 - \tau C\right)^2\left(G^2 + \left\|\nabla f(\theta)\right\|^2\right)$$

$\square$

**Lemma B.6** (Bounded Error of Momentum Update). *Consider the update rule in Eq. (5), and call $\tilde{g}^{t\tau} = \frac{1}{\tau} \sum_{k=t-\tau+1}^{t} \frac{1}{|\mathcal{S}^k|J} \sum_{i=1}^{|\mathcal{S}^k|} \sum_{j=1}^{J} \tilde{g}_i^{k,j}(\theta_i^{k,j-1})$ the server stochastic average pseudo-gradient over the last $\tau$ global steps and the average server pseudo-gradient at current parameters as $g^{t\tau} := {}^1/|\mathcal{S}_\tau^t| \sum_{i=1}^{|\mathcal{S}_\tau^t|} g_i^t(\theta^{t-1})$. Let also define the client drift $\mathcal{U}_t := \frac{1}{|\mathcal{S}|J} \sum_{j=1}^{J} \sum_{i=1}^{|\mathcal{S}|} \mathbb{E}\|\theta_i^{t,j} - \theta^{t-1}\|^2$ and the error of server update $\mathcal{E}_t := \mathbb{E}\|\nabla f(\theta^{t-1}) - \tilde{m}_\tau^{t+1}\|^2$. Under Assumptions 4.1, 4.2 and 4.4, it holds that:*

$$\mathbb{E}\left[\|\tilde{g}^{t\tau} - g^{t\tau}\|^2\right] \leq 3\left(\frac{\sigma^2}{|\mathcal{S}_\tau^t|J} + \frac{L^2}{\tau}\sum_{k=t-\tau+1}^{t} \mathcal{U}_k + 2L^2\eta^2 \sum_{k=t-\tau+1}^{t-1}\left(\mathbb{E}\left[\|\nabla f(\theta^{k-1})\|^2\right] + \mathcal{E}_k\right)\right)$$

**Proof of Lemma B.6** (Bounded error of delayed gradients)

Note that, by Assumption 4.4, $|\mathcal{S}^t| = |\mathcal{S}|C \,\forall t$, and that $|\mathcal{S}|C\tau = |\mathcal{S}_\tau^t|$:

$$\Lambda_t = \mathbb{E}\left[\left\|\frac{1}{\tau}\sum_{k=t-\tau+1}^{t}\frac{1}{|\mathcal{S}^k|J}\sum_{i=1}^{|\mathcal{S}^k|}\sum_{j=1}^{J}\tilde{g}_i^{k,j}(\theta_i^{k,j-1}) - g^{t\tau}\right\|^2\right] \tag{42}$$

$$= \mathbb{E}\left[\left\|\frac{1}{\tau}\sum_{k=t-\tau+1}^{t}\frac{1}{|\mathcal{S}^k|J}\sum_{i=1}^{|\mathcal{S}^k|}\sum_{j=1}^{J}\left(\tilde{g}_i^{k,j}(\theta_i^{k,j-1}) - g_i(\theta^{t-1})\right)\right\|^2\right] \tag{43}$$

$$= \mathbb{E}\left[\left\|\frac{1}{\tau}\sum_{k=t-\tau+1}^{t}\frac{1}{|\mathcal{S}^k|J}\sum_{i=1}^{|\mathcal{S}^k|}\sum_{j=1}^{J}\left(\tilde{g}_i^{k,j}(\theta_i^{k,j-1}) - g_i(\theta_i^{k,j-1}) + g_i(\theta_i^{k,j-1}) - g_i(\theta^{k-1}) + g_i(\theta^{k-1}) - g_i(\theta^{t-1})\right)\right\|^2\right] \tag{44}$$

$$\leq 3\left(\mathcal{T}_1 + \mathcal{T}_2 + \mathcal{T}_3\right)$$

$$\mathcal{T}_1 = \mathbb{E}\left[\left\|\frac{1}{\tau}\sum_{k=t-\tau+1}^{t}\frac{1}{|\mathcal{S}^k|J}\sum_{i=1}^{|\mathcal{S}^k|}\sum_{j=1}^{J}\left(\tilde{g}_i^{k,j}(\theta_i^{k,j-1}) - g_i(\theta_i^{k,j-1})\right)\right\|^2\right]$$

$$\leq \frac{1}{\tau}\frac{\sigma^2}{|\mathcal{S}^t|J} = \frac{\sigma^2}{|\mathcal{S}_\tau^t|J} \tag{45}$$

$$\mathcal{T}_2 = \mathbb{E}\left[\left\|\frac{1}{\tau}\sum_{k=t-\tau+1}^{t}\frac{1}{|\mathcal{S}^k|J}\sum_{i=1}^{|\mathcal{S}^k|}\sum_{j=1}^{J}\left(g_i(\theta_i^{k,j-1}) - g_i(\theta^{k-1})\right)\right\|^2\right] \tag{46}$$

$$\leq \frac{L^2}{|\mathcal{S}|J\tau}\sum_{k=t-\tau+1}^{t}\sum_{i=1}^{|\mathcal{S}|}\sum_{j=1}^{J}\mathbb{E}\left[\|\theta^{k,j-1} - \theta^{k-1}\|^2\right] \tag{47}$$

$$= \frac{L^2}{\tau}\sum_{k=t-\tau+1}^{t}\mathcal{U}_k \tag{48}$$

$$\tag{49}$$

$$T_3 = \mathbb{E}\left[\left\|\frac{1}{\tau}\sum_{k=t-\tau+1}^{t}\frac{1}{|\mathcal{S}^k|J}\sum_{i=1}^{|\mathcal{S}^k|}\sum_{j=1}^{J}\left(g_i(\theta^{k-1}) - g_i(\theta^{t-1})\right)\right\|^2\right] \tag{50}$$

$$\leq \frac{L^2}{|\mathcal{S}|\tau}\sum_{k=t-\tau+1}^{t}\sum_{i=1}^{|\mathcal{S}|}\mathbb{E}\left[\left\|\theta^{k-1} - \theta^{t-1}\right\|^2\right] \tag{51}$$

$$\leq \frac{L^2}{\tau}\sum_{k=t-\tau+1}^{t}\mathbb{E}\left[\left\|\theta^{k-1} - \theta^{t-1}\right\|^2\right] \tag{52}$$

$$= \frac{L^2}{\tau}\sum_{k=t-\tau+1}^{t}(t-k)\,\mathbb{E}\left[\left\|\theta^{k} - \theta^{k-1}\right\|^2\right] \tag{53}$$

$$\leq 2L^2\eta^2\sum_{k=t-\tau+1}^{t-1}\left(\mathbb{E}\left[\left\|\nabla f(\theta^{k-1})\right\|^2\right] + \mathcal{E}_k\right) \tag{54}$$

So, combining with lemma Lemmas B.8 and B.9 we have:

$$\sum_{t=1}^{T}\Lambda_t \leq 3\left(\frac{T\sigma^2}{|\mathcal{S}_\tau^t|J} + L^2\sum_{t=1}^{T}\mathcal{U}_t + 2L^2\eta^2(\tau-1)\sum_{t=1}^{T-1}\left(\mathbb{E}\left[\left\|\nabla f(\theta^{t-1})\right\|^2\right] + \mathcal{E}_t\right)\right) \tag{55}$$

$$\overset{\text{lemma B.8}}{=} 3\left(\frac{T\sigma^2}{|\mathcal{S}_\tau^t|J} + 2L^2\eta^2(\tau-1)\sum_{t=1}^{T-1}\left(\mathbb{E}\left[\left\|\nabla f(\theta^{t-1})\right\|^2\right] + \mathcal{E}_t\right)\right. \tag{56}$$

$$\left. + \underbrace{L^2TJ\eta_l^2\beta^2\sigma^2\left(1 + 2J^3\eta_l^2\beta^2L^2\right)}_{T_4} + 2J^2L^2e^2\sum_{t=1}^{T}\Xi_t)\right)$$

$$\overset{\text{lemma B.9}}{=} 3\left(\frac{T\sigma^2}{|\mathcal{S}_\tau^t|J} + 2L^2\eta^2(\tau-1)\sum_{t=1}^{T-1}\left(\mathbb{E}\left[\left\|\nabla f(\theta^{t-1})\right\|^2\right] + \mathcal{E}_t\right)\right. \tag{57}$$

$$+ T_4 + \underbrace{2J^2L^2e^2\left(4\eta_l^2\left((1-\beta)^2 + e(\beta\eta LT)^2\right)\right)}_{\alpha_1}\sum_{t=0}^{T-1}\left(\mathcal{E}_t + \mathbb{E}\left[\left\|\nabla f(\theta^{t-1})\right\|^2\right]\right)$$

$$\left. + \underbrace{2e^2J^2L^2(2e\eta_l^2\beta\tau TG_\tau)}_{T_5}\right)$$

$$= 3\left(\frac{T\sigma^2}{|\mathcal{S}_\tau^t|J} + T_4 + \underbrace{(\alpha_1 + 2L^2\eta_l^2(\tau-1))}_{\alpha_2}\sum_{t=1}^{T-1}\left(\mathbb{E}\left[\left\|\nabla f(\theta^{t-1})\right\|^2\right] + \mathcal{E}_t\right) + T_5\right) \tag{58}$$

$$\square$$

## B.4 Convergence Proof

**Lemma B.7** (Bounded variance of server updates). *Under Assumptions 4.1 and 4.2, it holds that:*

$$\sum_{t=1}^{T}\mathcal{E}_t \leq \frac{8}{5\beta}\mathcal{E}_0 + \frac{3}{5}\sum_{t=0}^{T-1}\mathbb{E}\left[\left\|\nabla f(\theta^{t-1})\right\|^2\right] + 21\beta\frac{\sigma^2}{|\mathcal{S}_\tau^t|J}T + \tag{59}$$

$$+ \frac{448}{5}(\eta_l JL)^2(e^3\tau T)G_\tau + 6\beta\sum_{t=1}^{T}\gamma_t$$

*Proof.*

$$\mathcal{E}_t := \mathbb{E}\left[\left\|\nabla f(\theta^{t-1}) - \tilde{m}_\tau^{t+1}\right\|^2\right] \tag{60}$$

$$= \mathbb{E}\left[\left\|(1-\beta)(\nabla f(\theta^{t-1}) - \tilde{m}_\tau^t) + \beta(\nabla f(\theta^{t-1}) - \tilde{g}^{t_\tau})\right\|^2\right] \tag{61}$$

$$= \mathbb{E}\left[\left\|(1-\beta)(\nabla f(\theta^{t-1}) - \tilde{m}_\tau^t)\right\|^2\right] + \beta^2\mathbb{E}\left[\left\|(\nabla f(\theta^{t-1}) - \tilde{g}^{t_\tau})\right\|^2\right] \tag{62}$$

$$+ 2\beta\mathbb{E}\left[\left\langle (1-\beta)(\nabla f(\theta^{t-1}) - \tilde{m}_\tau^t), \nabla f(\theta^{t-1}) - \frac{1}{\tau}\sum_{k=t-\tau+1}^{t} \frac{1}{|\mathcal{S}^k|J}\sum_{i=1}^{|\mathcal{S}^k|}\sum_{j=1}^{J} g_i(\theta_i^{k,j-1})\right\rangle\right] \tag{63}$$

Using the AM-GM inequality and Lemma B.3:

$$\leq \left(1+\frac{\beta}{2}\right)\mathbb{E}\left[\left\|(1-\beta)(\nabla f(\theta^{t-1}) - \tilde{m}_\tau^t)\right\|^2\right] + 2\beta^2\left(\gamma_t + \Lambda_t\right) +$$

$$+ 4\beta\gamma_t + 8\beta\left(\frac{L^2}{\tau}\sum_{k=t-\tau+1}^{t}\mathcal{U}_k + 2L^2\eta^2\sum_{k=t-\tau+1}^{t-1}\left(\mathbb{E}\left[\left\|\nabla f(\theta^{k-1})\right\|^2\right] + \mathcal{E}_k\right)\right) \tag{64}$$

$$\overset{\text{lemma B.6}}{\leq} \left(1+\frac{\beta}{2}\right)\mathbb{E}\left[\left\|(1-\beta)(\nabla f(\theta^{t-1}) - \tilde{m}_\tau^t)\right\|^2\right] + \left(2\beta^2 + 4\beta\right)\gamma_t + 6\beta^2\frac{\sigma^2}{|\mathcal{S}_\tau^t|J} + \tag{65}$$

$$+ \left(6\beta^2 + 8\beta\right)\underbrace{\left(\frac{L^2}{\tau}\sum_{k=t-\tau+1}^{t}\mathcal{U}_k + 2L^2\eta^2\sum_{k=t-\tau+1}^{t-1}\left(\mathbb{E}\left[\left\|\nabla f(\theta^{k-1})\right\|^2\right] + \mathcal{E}_k\right)\right)}_{\mathcal{T}_1}$$

$$\leq (1-\beta)^2\left(1+\frac{\beta}{2}\right)\mathbb{E}\left[\left\|\nabla f(\theta^{t-2}) - \tilde{m}_\tau^t + \nabla f(\theta^{t-1}) - \nabla f(\theta^{t-2})\right\|^2\right] + \tag{66}$$

$$+ 6\beta^2\frac{\sigma^2}{|\mathcal{S}_\tau^t|J} + 6\beta\gamma_t + 14\beta\mathcal{T}_1$$

Applying the AM-GM inequality again:

$$\leq (1-\beta)^2\left(1+\frac{\beta}{2}\right)\left[\left(1+\frac{\beta}{4}\right)\mathbb{E}\left[\left\|\nabla f(\theta^{t-2}) - \tilde{m}_\tau^t\right\|^2\right] + \tag{67}$$

$$+ \left(1+\frac{1}{\beta}\right)\mathbb{E}\left[\left\|\nabla f(\theta^{t-1}) - \nabla f(\theta^{t-2})\right\|^2\right]\right] + 6\beta^2\frac{\sigma^2}{|\mathcal{S}_\tau^t|J} + 6\beta\gamma_t + 14\beta\mathcal{T}_1$$

$$\overset{\text{assumption 4.2}}{\leq} (1-\beta)^2\left(1+\frac{\beta}{2}\right)\left[\left(1+\frac{\beta}{4}\right)\mathcal{E}_{t-1} + \tag{68}$$

$$+ \left(1+\frac{1}{\beta}\right)L^2\mathbb{E}\left[\left\|\theta^{t-1} - \theta^{t-2}\right\|^2\right]\right] + 6\beta^2\frac{\sigma^2}{|\mathcal{S}_\tau^t|J} + 6\beta\gamma_t + 14\beta\mathcal{T}_1$$

$$\leq (1-\beta)^2\left(1+\frac{\beta}{2}\right)\left[\left(1+\frac{\beta}{4}\right)\mathcal{E}_{t-1} + \tag{69}$$

$$+ 2\left(1+\frac{1}{\beta}\right)L^2\eta^2\left(\mathbb{E}\left[\left\|\nabla f(\theta^{t-2})\right\|^2\right] + \mathcal{E}_{t-1}\right)\right] + 6\beta^2\frac{\sigma^2}{|\mathcal{S}_\tau^t|J} + 6\beta\gamma_t + 14\beta\mathcal{T}_1$$

Where in the last inequality we used the fact that:

$$\left\|\theta^{t-1} - \theta^{t-2}\right\|^2 \leq 2\eta^2\left(\left\|\nabla f(\theta^{t-2})\right\|^2 + \left\|\nabla f(\theta^{t-2}) - \tilde{m}_\tau^t\right\|^2\right).$$

Now notice that $(1-\beta)^2\left(1+\frac{\beta}{2}\right)\left(1+\frac{\beta}{4}\right) \leq (1-\beta)$ and that $2(1-\beta)^2\left(1+\frac{\beta}{2}\right)\left(1+\frac{1}{\beta}\right) \leq \frac{2}{\beta}$:

$$\mathcal{E}_t \leq (1-\beta)\mathcal{E}_{t-1} + \frac{2}{\beta}L^2\eta^2\left(\mathbb{E}\left[\left\|\nabla f(\theta^{t-2})\right\|^2\right] + \mathcal{E}_{t-1}\right) + 6\beta^2\frac{\sigma^2}{|\mathcal{S}_\tau^t|J} + 6\beta\gamma_t + 14\beta\mathcal{T}_1 \tag{70}$$

$$= \left(1-\beta+\frac{2}{\beta}L^2\eta^2\right)\mathcal{E}_{t-1} + \frac{2}{\beta}L^2\eta^2\mathbb{E}\left[\left\|\nabla f(\theta^{t-2})\right\|^2\right] + 6\beta^2\frac{\sigma^2}{|\mathcal{S}_\tau^t|J} + 6\beta\gamma_t + 14\beta\mathcal{T}_1 \tag{71}$$

Define:

- $\mathcal{T}_2 := L^2 T J \eta_l^2 \beta^2 \sigma^2 \left(1 + 2J^3 \eta_l^2 \beta^2 L^2\right)$

- $\mathcal{T}_3 := 2e^2 J^2 L^2 (2e\eta_l^2 \beta \tau T G_\tau)$

- $\alpha_1 := 2J^2 L^2 e^2 \left(4\eta_l^2 \left((1-\beta)^2 + e(\beta\eta LT)^2\right)\right) + 2L^2 \eta_l^2 (\tau - 1)$

Summing up over $T$ and substituting into $\mathcal{T}_1$ the expression for $\mathcal{U}_t$:

$$\sum_{t=1}^{T} \mathcal{E}_t \leq \underbrace{\left(1 - \beta + \frac{2}{\beta} L^2 \eta^2 + 14\beta\alpha_1\right)}_{\alpha_2} \sum_{t=0}^{T-1} \mathcal{E}_t + \tag{72}$$

$$+ \underbrace{\left(\frac{2}{\beta} L^2 \eta^2 + 14\beta\alpha_1\right)}_{\alpha_3} \sum_{t=0}^{T-1} \mathbb{E}\left[\left\|\nabla f(\theta^{t-1})\right\|^2\right] +$$

$$+ 14\beta \left(\mathcal{T}_2 + \mathcal{T}_3\right) T + 6\beta^2 \frac{\sigma^2}{|\mathcal{S}_\tau^t| J} T + 6\beta \sum_{t=1}^{T} \gamma_t$$

We now have that:

$$\alpha_2 := \left(1 - \beta + \frac{2}{\beta} L^2 \eta^2 + 14\beta \left[2J^2 L^2 e^2 \left(4\eta_l^2 \left((1-\beta)^2 + e(\beta\eta LT)^2\right)\right) + 2L^2 \eta_l^2 (\tau - 1)\right]\right) \tag{73}$$

$$= \left(1 - \beta + \frac{2}{\beta} L^2 \eta^2 + 14\beta \left[8J^2 L^2 e^2 \eta_l^2 \left((1-\beta)^2 + e(\beta\eta LT)^2\right) + 2L^2 \eta_l^2 (\tau - 1)\right]\right) \tag{74}$$

$$\leq \left(1 - \beta + \frac{2}{\beta} L^2 \eta^2 + 112\beta e^2 (\eta_l J L)^2 \left[(1-\beta)^2 + (\beta\eta LT)^2 + (\tau - 1)\right]\right) \tag{75}$$

$$\tag{76}$$

Now impose $(\eta_l J L) \leq (37\sqrt{\tau}\beta\eta LT e)^{-1}$ and $\eta \leq \frac{\beta}{\sqrt{8}L}$. We have that:

$$\alpha_2 \leq \left(1 - \beta + \frac{2\beta}{8} + \frac{\beta}{8}\right) = \left(1 - \frac{5\beta}{8}\right) \tag{77}$$

$$\alpha_3 \leq \frac{3\beta}{8} \tag{78}$$

$$14\beta\mathcal{T}_2 = 14\beta L^2 T J \eta_l^2 \beta^2 \sigma^2 \left(1 + 2J^3 \eta_l^2 \beta^2 L^2\right) \tag{79}$$

$$= 14\beta^3 (\eta_l J L)^2 \left(\frac{1}{J} + 2(\eta_l J L \beta)^2\right) \sigma^2 T \tag{80}$$

$$\leq 7\beta^2 \frac{\sigma^2}{|\mathcal{S}_\tau^t| J} T \tag{81}$$

Where in the last inequality we apply:

$$2\beta(\eta_l J L)^2 \left(\frac{1}{J} + 2(\eta_l J L \beta)^2\right) \leq \frac{1}{|\mathcal{S}_\tau^t| J}$$

Plugging all the terms together we have:

$$\sum_{t=1}^{T} \mathcal{E}_t \leq \left(1 - \frac{5}{8\beta}\right) \sum_{t=0}^{T-1} \mathcal{E}_t + \frac{3\beta}{8} \sum_{t=0}^{T-1} \mathbb{E}\left[\left\|\nabla f(\theta^{t-1})\right\|^2\right] + 13\beta^2 \frac{\sigma^2}{|\mathcal{S}_\tau^t| J} T + \tag{82}$$

$$+ 56\beta(\eta_l J L)^2 (e^3 \tau T) G_\tau + 6\beta \sum_{t=1}^{T} \gamma_t$$

Rearranging the terms completes the proof. $\qquad\square$

**Lemma B.8.** *Under Assumptions 4.1 and 4.2, for Eq. (9) it holds that:*

$$\mathcal{U}_t \le 2J^2 e^2 \Xi_t + J\eta_l^2 \beta^2 \sigma^2 (1 + 2J^3 \eta_l^2 L^2 \beta^2) \tag{83}$$

$$\sum_{t=1}^{T} \mathcal{U}_t \le TJ\eta_l^2 \beta^2 \sigma^2 (1 + 2J^3 \eta_l^2 \beta^2 L^2) + 2J^2 e^2 \sum_{t=1}^{T} \Xi_t \tag{84}$$

*Proof.*

$$\mathbb{E}\left[\left\|\theta_i^{t,j} - \theta^{t-1}\right\|^2\right] \le 2\mathbb{E}\left[\left\|\sum_{k=0}^{j-1} \zeta_i^{t,k}\right\|^2\right] + 2j\eta_l^2 \beta^2 \sigma^2 \tag{85}$$

$$\stackrel{\text{lemma B.3}}{\le} 2j \sum_{k=0}^{j-1} \mathbb{E}\left[\left\|\zeta_i^{t,k}\right\|^2\right] + 2j\eta_l^2 \beta^2 \sigma^2 \tag{86}$$

For any $1 \le k \le j - 1 \le J - 2$, using $\eta L \le \frac{1}{\beta J} \le \frac{1}{\beta(j+1)}$, we have:

$$\mathbb{E}\left[\left\|\zeta_i^{t,k}\right\|^2\right] \le \left(1 + \frac{1}{j}\right)\mathbb{E}\left[\left\|\zeta_i^{t,k-1}\right\|^2\right] + (1+j)\mathbb{E}\left[\left\|\zeta_i^{t,k} - \zeta_i^{t,k-1}\right\|^2\right] \tag{87}$$

$$\le \left(1 + \frac{1}{j}\right)\mathbb{E}\left[\left\|\zeta_i^{t,k-1}\right\|^2\right] + (1+j)\eta_l^2 \beta^2 L^2 \left(\eta_l^2 \beta^2 \sigma^2 + \mathbb{E}\left[\left\|\zeta_i^{t,k-1}\right\|^2\right]\right) \tag{88}$$

$$\le \left(1 + \frac{1}{j}\right)\mathbb{E}\left[\left\|\zeta_i^{t,k-1}\right\|^2\right] + (1+j)\eta_l^4 \beta^4 L^2 \sigma^2 + \frac{1}{1+j}\mathbb{E}\left[\left\|\zeta_i^{t,k} - \zeta_i^{t,k-1}\right\|^2\right] \tag{89}$$

$$\le \left(1 + \frac{2}{j}\right)\mathbb{E}\left[\left\|\zeta_i^{t,k-1}\right\|^2\right] + (1+j)\eta_l^4 \beta^4 L^2 \sigma^2 \tag{90}$$

$$\stackrel{\left(1+\frac{2}{j}\right)^j \le e^2}{\le} e^2 \mathbb{E}\left[\left\|\zeta_i^{t,0}\right\|^2\right] + 4j^2 \eta_l^4 \beta^4 L^2 \sigma^2 \tag{91}$$

So it holds that:

$$\mathbb{E}\left[\left\|\theta_i^{t,j} - \theta^{t-1}\right\|^2\right] \le 2j^2\left(e^2 \mathbb{E}\left[\left\|\zeta_i^{t,0}\right\|^2\right] + 4j^2 \eta_l^4 L^2 \sigma^2\right) + 2j\eta_l^2 \sigma^2 \tag{92}$$

$$= 2e^2 j^2 \mathbb{E}\left[\left\|\zeta_i^{t,0}\right\|^2\right] + 2j\eta_l^2 \sigma^2 \beta^2 (1 + 4j^3 \eta_l^2 L^2 \beta^2) \tag{93}$$

So, summing up over $i$ and $j$:

$$\mathcal{U}_t \le \frac{1}{|\mathcal{S}|J} \sum_{i=1}^{|\mathcal{S}|} \sum_{j=1}^{J} 2e^2 j^2 \mathbb{E}\left[\left\|\zeta_i^{t,0}\right\|^2\right] + 2j\eta_l^2 \sigma^2 \beta^2 (1 + 4j^3 \eta_l^2 L^2 \beta^2) \tag{94}$$

$$\le 2J^2 e^2 \Xi_t + J\eta_l^2 \beta^2 \sigma^2 (1 + 2J^3 \eta_l^2 L^2 \beta^2) \tag{95}$$

Finally, summing up over $T$:

$$\sum_{t=1}^{T} \mathcal{U}_t \le \underbrace{TJ\eta_l^2 \beta^2 \sigma^2 (1 + 2J^3 \eta_l^2 \beta^2 L^2)}_{\mathcal{T}_1} + 2J^2 e^2 \sum_{t=1}^{T} \Xi_t \tag{96}$$

$$\le \mathcal{T}_1 + 2J^2 e^2 \left(4\eta^2\left((1-\beta)^2 + e(\beta\eta LT)^2\right) \sum_{t=1}^{T-1}\left(\mathcal{E}_t + \mathbb{E}\left[\left\|\nabla f(\theta^{t-1})\right\|^2\right]\right) + \underbrace{2e\eta^2 \beta^2 \tau TG_\tau}_{\mathcal{T}_2}\right) \tag{97}$$

$$\le \mathcal{T}_1 + \alpha_1 \sum_{t=1}^{T-1}\left(\mathcal{E}_t + \mathbb{E}\left[\left\|\nabla f(\theta^{t-1})\right\|^2\right]\right) + \alpha_2 \mathcal{T}_2 \tag{98}$$

$\square$

**Lemma B.9.** *Under Assumptions 4.1, 4.2 and 4.4, if $224e(\eta_l JL)^2\left((1-\beta)^2 + e(\beta\eta LT)^2\right) \le 1$, for Eq. (11) it holds for $t \ge 0$ that:*

$$\Xi_t \le \frac{1}{56eJ^2L^2}\sum_{t=0}^{T-1}\left(\mathcal{E}_t + \mathbb{E}\left[\left\|\nabla f(\theta^{t-1})\right\|^2\right]\right) + 2e\eta_l^2\beta^2\tau T G_\tau \tag{99}$$

*Proof.* Note that $\zeta_i^{t,0} = -\eta_l\left((1-\beta)\tilde{m}_\tau^t + \beta g_i(\theta^{t-1})\right)$,

$$\frac{1}{|\mathcal{S}|}\sum_{i=1}^{|\mathcal{S}|}\left\|\zeta_i^{t,0}\right\|^2 \le 2\eta_l^2\left((1-\beta)^2\left\|\tilde{m}_\tau^t\right\|^2 + \frac{\beta^2}{|\mathcal{S}|}\sum_{i=1}^{|\mathcal{S}|}\left\|g_i(\theta^{t-1})\right\|^2\right) \tag{100}$$

For any $a > 0$, considering each client participates to the train every $\tau = \frac{1}{C}$ rounds:

$$\mathbb{E}\left[\left\|g_i(\theta^{t-1})\right\|^2\right] = \mathbb{E}\left[\left\|g_i(\theta^{t-1}) - g_i(\theta^{t-\tau-1}) + g_i(\theta^{t-\tau-1})\right\|^2\right] \tag{101}$$

$$\overset{\text{lemma B.3}}{\le} (1+a)\mathbb{E}\left[\left\|g_i(\theta^{t-\tau-1})\right\|^2\right] + \tag{102}$$

$$+ \left(1+\frac{1}{a}\right)\mathbb{E}\left[\left\|g_i(\theta^{t-1}) - g_i(\theta^{t-\tau-1})\right\|^2\right]$$

$$\le (1+a)\mathbb{E}\left[\left\|g_i(\theta^{t-\tau-1})\right\|^2\right] + \tag{103}$$

$$+ \left(1+\frac{1}{a}\right)L^2\mathbb{E}\left[\left\|\theta^{t-1} - \theta^{t-\tau-1}\right\|^2\right] \tag{104}$$

$$\le (1+a)\mathbb{E}\left[\left\|g_i(\theta^{t-\tau-1})\right\|^2\right] + \tag{105}$$

$$+ 2\left(1+\frac{1}{a}\right)L^2\eta^2\tau\sum_{k=1}^{\tau}\left(\mathcal{E}_{t-k} + \mathbb{E}\left[\left\|\nabla f(\theta^{t-k-1})\right\|^2\right]\right) \tag{106}$$

$$\le (1+a)^{\frac{t}{\tau}}\mathbb{E}\left[\left\|g_i(\theta^{t_i-1})\right\|^2\right] + \tag{107}$$

$$+ 2\left(1+\frac{1}{a}\right)L^2\eta^2\tau\sum_{s=1}^{\frac{t}{\tau}}\sum_{k=1}^{\tau}\left(\mathcal{E}_{s\tau-k} + \mathbb{E}\left[\left\|\nabla f(\theta^{s\tau-k})\right\|^2\right]\right)(1+a)^{\frac{t}{\tau}-s}$$

$$\le (1+a)^{\frac{t}{\tau}}\mathbb{E}\left[\left\|g_i(\theta^{t_i-1})\right\|^2\right] + \tag{108}$$

$$+ 2\left(1+\frac{1}{a}\right)L^2\eta^2\tau\sum_{k=1}^{t-1}\left(\mathcal{E}_k + \mathbb{E}\left[\left\|\nabla f(\theta^{k-1})\right\|^2\right]\right)(1+a)^{\frac{t}{\tau}}$$

Where $t_i := \min_{t\in[T]}(t\ s.t.\ i\in\mathcal{S}^t)$. Now take $a = \frac{\tau}{t}$:

$$\mathbb{E}\left[\left\|g_i(\theta^{t-1})\right\|^2\right] \le e\mathbb{E}\left[\left\|g_i(\theta^{t_i-1})\right\|^2\right] + \tag{109}$$

$$+ 2e\eta^2 L^2\tau\left(\frac{t}{\tau}+1\right)\sum_{k=1}^{t-1}\left(\mathcal{E}_k + \mathbb{E}\left[\left\|\nabla f(\theta^{k-1})\right\|^2\right]\right)$$

So:

$$\sum_{t=1}^{T} \Xi_t \leq \sum_{t=1}^{T} 2\eta_l^2 \left( 2(1-\beta)^2 \left( \mathcal{E}_{t-1} + \mathbb{E}\left[ \left\| \nabla f(\theta^{t-2}) \right\|^2 \right] \right) + \frac{\beta^2}{|\mathcal{S}|} \sum_{i=1}^{|\mathcal{S}|} \mathbb{E}\left[ \left\| g_i(\theta^{t-1}) \right\|^2 \right] \right) \quad (110)$$

$$\leq \sum_{t=1}^{T} 4\eta_l^2 (1-\beta)^2 \left( \mathcal{E}_{t-1} + \mathbb{E}\left[ \left\| \nabla f(\theta^{t-2}) \right\|^2 \right] \right) + \quad (111)$$

$$+ 2\eta_l^2 \beta^2 \sum_{t=1}^{T} \left( \frac{e}{|\mathcal{S}|} \sum_{i=1}^{|\mathcal{S}|} \mathbb{E}\left[ \left\| g_i(\theta^{t_i-1}) \right\|^2 \right] + 2e\eta_l^2 L^2 \tau \left( \frac{t}{\tau} + 1 \right) \sum_{k=1}^{t-1} \left( \mathcal{E}_k + \mathbb{E}\left[ \left\| \nabla f(\theta^{t-1}) \right\|^2 \right] \right) \right)$$

$$\leq 4\eta_l^2 (1-\beta)^2 \sum_{t=1}^{T} \left( \mathcal{E}_{t-1} + \mathbb{E}\left[ \left\| \nabla f(\theta^{t-2}) \right\|^2 \right] \right) + \quad (112)$$

$$+ 2\eta_l^2 \beta^2 \left( eT \sum_{t=1}^{\tau} G_t + 2e(\eta L T)^2 \sum_{t=1}^{T-1} \left( \mathcal{E}_t + \mathbb{E}\left[ \left\| \nabla f(\theta^{t-1}) \right\|^2 \right] \right) \right)$$

Let us define $G_\tau := \max_{t \in [1,\tau]} G_t$, with $G_t := \frac{1}{|\mathcal{S}^t|} \sum_{i=1}^{|\mathcal{S}^t|} \mathbb{E}\left[ \left\| g_i(\theta^{t-1}) \right\|^2 \right]$. We have that:

$$\sum_{t=1}^{T} \Xi_t \leq 4\eta_l^2 \left( (1-\beta)^2 + e(\beta \eta L T)^2 \right) \sum_{t=0}^{T-1} \left( \mathcal{E}_t + \mathbb{E}\left[ \left\| \nabla f(\theta^{t-1}) \right\|^2 \right] \right) + 2e\eta_l^2 \beta^2 \tau T G_\tau \quad (113)$$

Applying the upper bound of $\eta_l$ completes the proof. $\qquad \square$

**Lemma B.10** (Cheng et al. [2024]). *Under Assumption 4.2, if $\eta L \leq \frac{1}{24}$, the following holds for all $t \geq 0$:*

$$\mathbb{E}\left[ f(\theta^t) \right] \leq \mathbb{E}\left[ f(\theta^{t-1}) \right] - \frac{11\eta}{24} \mathbb{E}\left[ \left\| \nabla f(\theta^{t-1}) \right\|^2 \right] + \frac{13\eta}{24} \mathcal{E}_t \quad (114)$$

*Proof.* Since f is $L$-smooth, we have:

$$f(\theta^t) \leq f(\theta^{t-1}) + \left\langle \nabla f(\theta^{t-1}), \theta^t - \theta^{t-1} \right\rangle + \frac{L}{2} \left\| \theta^t - \theta^{t-1} \right\|^2 \quad (115)$$

$$= f(\theta^{t-1}) - \eta \left\| \nabla f(\theta^{t-1}) \right\|^2 + \eta \left\langle \nabla f(\theta^{t-1}), \nabla f(\theta^{t-1}) - \tilde{m}_\tau^{t+1} \right\rangle + \frac{L\eta^2}{2} \left\| \tilde{m}_\tau^{t+1} \right\|^2 \quad (116)$$

Since $\theta^t = \theta^{t-1} - \eta \tilde{m}_\tau^{t+1}$, using Young's inequality and imposing $\eta L \leq \frac{1}{24}$, we further have:

$$f(\theta^t) \leq f(\theta^{t-1}) - \frac{\eta}{2} \left\| \nabla f(\theta^{t-1}) \right\|^2 + \frac{\eta}{2} \left\| \nabla f(\theta^{t-1}) - \tilde{m}_\tau^{t+1} \right\|^2 + \quad (117)$$

$$+ L\eta^2 \left( \left\| \nabla f(\theta^{t-1}) \right\|^2 + \left\| \nabla f(\theta^{t-1}) - \tilde{m}_\tau^{t+1} \right\|^2 \right)$$

$$\leq f(\theta^{t-1}) - \frac{11\eta}{24} \left\| \nabla f(\theta^{t-1}) \right\|^2 + \frac{13\eta}{24} \left\| \nabla f(\theta^{t-1}) - \tilde{m}_\tau^{t+1} \right\|^2 \quad (118)$$

$$\square$$

**Proof of Theorem 4.6** (Convergence rate of GHBM for non-convex functions)

*Under Assumptions 4.1, 4.2 and 4.4, if we take:*

$$\tilde{m}_\tau^0 = 0, \qquad \beta = \min \left\{ 1, \sqrt{\frac{|\mathcal{S}| J L \Delta}{\sigma^2 T}} \right\}, \qquad \eta = \min \left\{ \frac{1}{24L}, \frac{\beta}{\sqrt{8}L} \right\} \quad (119)$$

$$\eta_l J L \lesssim \min \left\{ 1, \frac{1}{\beta \eta L \sqrt{\tau} T}, \sqrt{\frac{L\Delta}{\beta^3 \tau G_\tau T}}, \frac{1}{\sqrt{\beta |\mathcal{S}|}}, \left( \frac{1}{\beta^3 |\mathcal{S}| J} \right)^{\frac{1}{4}} \right\}$$

*then GHBM with optimal $\tau = \frac{1}{C}$ converges as:*

$$\frac{1}{T} \sum_{t=1}^{T} \mathbb{E}\left[ \left\| \nabla f(\theta^{t-1}) \right\|^2 \right] \lesssim \frac{L\Delta}{T} + \sqrt{\frac{L\Delta\sigma^2}{|\mathcal{S}| J T}} \quad (120)$$

*Proof.* Combining the results of Lemmas B.7 and B.10, we have that:

$$\sum_{t=1}^{T} \left( \mathbb{E}\left[ f(\theta^t) \right] - \mathbb{E}\left[ f(\theta^{t-1}) \right] \right) \leq -\frac{11\eta}{24} \sum_{t=1}^{T} \mathbb{E}\left[ \left\| \nabla f(\theta^{t-1}) \right\|^2 \right] + \frac{13\eta}{24} \sum_{t=1}^{T} \mathcal{E}_t \tag{121}$$

$$\frac{1}{\eta} \mathbb{E}\left[ f(\theta^{t-1}) - f(\theta^0) \right] \leq \frac{26}{30\beta} \mathcal{E}_0 - \frac{1}{15} \sum_{t=1}^{T} \mathbb{E}\left[ \left\| \nabla f(\theta^{t-1}) \right\|^2 \right] + 32\beta \frac{\sigma^2}{|\mathcal{S}_\tau^t| J} T + \tag{122}$$

$$+ \frac{448}{5} (\eta_l J L)^2 (e^3 \tau T) G_\tau + 6\beta \sum_{t=1}^{T} \gamma_t \tag{123}$$

Imposing $\tau = \frac{1}{C}$, by Corollary B.5 we have that $\gamma_t = 0$ and $\mathcal{S}_\tau^t = \mathcal{S} \ \forall t$. Also, noticing that $\tilde{m}_\tau^0 = 0$ implies $\mathcal{E}_0 \leq 2L \left( f(\theta^0) - f^* \right) = 2L\Delta$, we have that:

$$\frac{1}{T} \sum_{t=1}^{T} \mathbb{E}\left[ \left\| \nabla f(\theta^{t-1}) \right\|^2 \right] \lesssim \frac{L\Delta}{\eta L T} + \frac{\mathcal{E}_0}{\beta T} + (\eta_l J L \beta)^2 \tau G_\tau + \beta \frac{\sigma^2}{|\mathcal{S}| J} \tag{124}$$

$$\lesssim \frac{L\Delta}{T} + \frac{2L\Delta}{\beta T} + (\eta_l J L \beta)^2 \tau G_\tau + \beta \frac{\sigma^2}{|\mathcal{S}| J} \tag{125}$$

$$\lesssim \frac{L\Delta}{T} + \frac{2L\Delta}{\beta T} + \beta^2 \left( \frac{L\Delta}{\beta^3 \tau G_\tau T} \right) \tau G_\tau + \beta \frac{\sigma^2}{|\mathcal{S}| J} \tag{126}$$

$$\lesssim \frac{L\Delta}{T} + \frac{L\Delta}{\beta T} + \beta \frac{\sigma^2}{|\mathcal{S}| J} \tag{127}$$

$$\lesssim \frac{L\Delta}{T} + \sqrt{\frac{L\Delta\sigma^2}{|\mathcal{S}| J T}} \tag{128}$$

where the fourth inequality follows from applying the upper bound $\eta_l J L \leq \sqrt{\frac{L\Delta}{\beta^3 \tau G_\tau T}}$ on the third term of Eq. (125). $\qquad\square$

## C  Experimental Setting

### C.1  Datasets and Models

**CIFAR-10/100.**  We consider CIFAR-10 and CIFAR-100 to experiment with image classification tasks, each one respectively having 10 and 100 classes. For all methods, training images are preprocessed by applying random crops, followed by random horizontal flips. Both training and test images are finally normalized according to their mean and standard deviation. As the main model for experimentation, we used a model similar to LENET-5 as proposed in [Hsu et al., 2020]. To further validate our findings, we also employed a RESNET-20 as described in [He et al., 2015], following the implementation provided in [Idelbayev, 2021]. Since batch normalization Ioffe and Szegedy [2015] layers have been shown to hamper performance in learning from decentralized data with skewed label distribution [Hsieh et al., 2020], we replaced them with group normalization [Wu and He, 2018], using two groups in each layer. For a fair comparison, we used the same modified network also in centralized training. We report the result of centralized training for reference in Table 5: as per the hyperparameters, we use 64 for the batch size, 0.01 and 0.1 for the learning rate respectively for the LENET and the RESNET-20 and 0.9 for momentum. We trained both models on both datasets for 150 epochs using a cosine annealing learning rate scheduler.

**Shakespeare.**  The Shakespeare language modeling dataset is created by collating the collective works of William Shakespeare and originally comprises 715 clients, with each client denoting a speaking role. However, for this study, a different approach was used, adopting the LEAF [Caldas et al., 2019] framework to split the dataset among 100 devices and restrict the number of data points per device to 2000. The non-IID dataset is formed by assigning each device to a specific role, and the local dataset for each device contains the sentences from that role. Conversely, the IID dataset is created by randomly distributing sentences from all roles across the devices.

Table 5: **Test accuracy (%) of centralized training over datasets and models used.** Results are reported in term of mean top-1 accuracy over the last 10 epochs, averaged over 5 independent runs.

| DATASET | ACC. CENTRALIZED (%) |
|---|---|
| CIFAR-10 W/ LENET | $86.48 \pm 0.22$ |
| CIFAR-10 W/ RESNET-20 | $89.05 \pm 0.44$ |
| CIFAR-100 W/ LENET | $57.00 \pm 0.09$ |
| CIFAR-100 W/ RESNET-20 | $62.21 \pm 0.85$ |
| SHAKESPEARE | $52.00 \pm 0.16$ |
| STACKOVERFLOW | $28.50 \pm 0.25$ |
| GLDV2 | $74.03 \pm 0.15$ |

For this task, we have employed a two-layer Long Short-Term Memory (LSTM) classifier, consisting of 100 hidden units and an 8-dimensional embedding layer. Our objective is to predict the next character in a sequence, where there are a total of 80 possible character classes. The model takes in a sequence of 80 characters as input, and for each character, it learns an 8-dimensional representation. The final output of the model is a single character prediction for each training example, achieved through the use of 2 LSTM layers and a densely-connected layer followed by a softmax. This model architecture is the same used by [Li et al., 2020, Acar et al., 2021].

We report the result of centralized training for reference in Table 5: we train for 75 epochs with constant learning rate, using as hyperparameters 100 for the batch size, 1 for the learning rate, 0.0001 for the weight decay and no momentum.

**StackOverflow.**  The Stack Overflow dataset is a language modeling corpus that comprises questions and answers from the popular Q&A website, StackOverflow. Initially, the dataset consists of 342477 unique users but for, practical reasons, we limit our analysis to a subset of $40k$ users. Our goal is to perform the next-word prediction on these text sequences. To achieve this, we utilize a Recurrent Neural Network (RNN) that first learns a 96-dimensional representation for each word in a sentence and then processes them through a single LSTM layer with a hidden dimension of 670. Finally, the model generates predictions using a densely connected softmax output layer. The model and the preprocessing steps are the same as in [Reddi et al., 2021]. We report the result of centralized training for reference in Table 5: as per the hyperparameters, we use 16 for the batch size, $10^{-1/2}$ for the learning rate and no momentum or weight decay. We train for 50 epochs with a constant learning rate. Given the size of the test dataset, testing is conducted on a subset of them made by 10000 randomly chosen test examples, selected at the beginning of training.

**Large-scale Real-world Datasets.**  As large-scale real-world datasets for our experimentation, we follow Hsu et al. [2020]. GLDv2 is composed of $\approx 164k$ images belonging to $\approx 2000$ classes, realistically split among 1262 clients. INATURALIST is composed of $\approx 120k$ images belonging to

$\approx 1200$ classes, split among 9275 clients. These datasets are challenging to train not only because of their inherent complexity (size of images, number of classes) but also because usually at each round a very small portion of clients is selected. In particular, for GLDv2 we sample 10 clients per round, while for INATURALIST we experiment with different participation rates, sampling 10, 50, or 100 clients per round. In the main paper, we choose to report the participation rate instead of the number of sampled clients to better highlight that the tested scenarios are closer to a cross-device setting, which is the most challenging for algorithms based on client participation, like SCAFFOLD and ours. As per the model, for both datasets, we use a MobileNetV2 pretrained on ImageNet.

**Details on the Experiment in Fig. 5.** In the main text (see Sec. 4.1) we provide an experiment to illustrate the convergence rate of GHBM (see Fig. 5). The learning problem consists in a linear regression of the coefficients $(a, b, c) \in \mathbb{R}$ of a quadratic function $f(x) = ax^2 + bx + c$. The synthetic dataset is made of 6400 observations of the above function (with $a = 10, b = 5, c = -1$) in the range $x \in [-10, 10]$. The dataset is split among $K = 50$ clients each one having 128 samples, and non-iidness is simulated by splitting the domain into equally big disjoint subsets, and having each client the observation of that domain.

Table 6: Details about datasets' split used for our experiments

|  | CIFAR-10 | CIFAR-100 | SHAKESPEARE | STACKOVERFLOW | GLDv2 | INATURALIST |
|---|---|---|---|---|---|---|
| Clients | 100 | 100 | 100 | 40.000 | 1262 | 9275 |
| Number of clients per round | 10 | 10 | 10 | 50 | 10 | $\{10, 50, 100\}$ |
| Number of classes | 10 | 100 | 80 | 10004 | 2028 | 1203 |
| Avg. examples per client | 500 | 500 | 2000 | 428 | 130 | 13 |
| Number of local steps | 8 | 8 | 20 | 27 | 13 | 2 |
| Average participation (round no.) | 1k | 1k | 25 | 1.5 | 40 | $\{5, 27, 54\}$ |

## C.2 Simulating Heterogeneity

For CIFAR-10/100 we simulate arbitrary heterogeneity by splitting the total datasets according to a Dirichlet distribution with concentration parameter $\alpha$, following Hsu et al. [2020]. In practice, we draw a multinomial $q_i \sim \mathbf{Dir}(\alpha p)$ from a Dirichlet distribution, where $p$ describes a prior class distribution over $N$ classes, and $\alpha$ controls the heterogeneity among all clients: the greater $\alpha$ the more homogeneous the clients' data distributions will be. After drawing the class distributions $q_i$, for every client $i$, we sample training examples for each class according to $q_i$ without replacement.

## C.3 Evaluating Communication and Computational Cost

In the main paper we showed a comparison in communication and computational cost of state-of-art FL algorithms compared to our solutions GHBM and FEDHBM: in this section we detail how those results in table Tab. 3 have been obtained. We follow a three-step procedure:

1. For each algorithm $a$, we calculate the minimum number of rounds $r_a$ to reach the performance of FEDAVG, the total amount of bytes exchanged $b_a$ in the whole training budget (number of rounds, as described in Appendix C.5) and the measure the corresponding total training time $t_a$. In this way, the different requirements in communication and computation of each algorithm are taken into account for the next steps.

2. We calculate the actual communication and computational requirements as $(tb_a = b_a \cdot s_a, tt_a = t_a \cdot s_a)$, where $s_a = \frac{r_a}{T}$ is the speedup of the algorithm w.r.t. FEDAVG. For those competitor algorithms that did not reach the target performance (*e.g.* MIMEMOM) in the training budget $T$, we conservatively consider $r_a = T$. In this way, the convergence speed of each algorithm is taken into account for determining the actual amount of computation needed.

3. We complement the above information with with a reduction/increase factor w.r.t. FEDAVG, calculated as $rtb_a = \left(1 - \frac{tb_a}{tb_{\text{FEDAVG}}}\right)$ and $rtt_a = \left(1 - \frac{tt_a}{tt_{\text{FEDAVG}}}\right)$ and expressed as a percentage. A cost reduction (*i.e.* $rtb_a > 0$ or $rtt_a > 0$) is indicated with $\downarrow$, while a cost increase (*i.e.* $rtb_a < 0$ or $rtt_a < 0$) is indicated with $\uparrow$. This gives a practical indication of how much communication/computation have been saved in choosing the algorithm at hand as an alternative for FEDAVG.

Table 7: Hyper-parameter search grid for each combination of method and dataset (for $\alpha = 0$). The best values are indicated in **bold**.

| METHOD | HPARAM | CIFAR-10/100 | | SHAKESPEARE | STACKOVERFLOW |
|---|---|---|---|---|---|
| | | LENET | RESNET-20 | | |
| ALL FL | wd | [**0.001**, 0.0008, 0.0004] | [0.0001, **0.00001**] | [0, **0.0001**, 0.00001] | [**0**, 0.0001, 0.00001] |
| | $B$ | 64 | 64 | 100 | 16 |
| FEDAVG | $\eta$ | [2, **1.5**, 1, 0.5, 0.1] | [1.5, **1**, 0.1] | [1.5, **1**, 0.5, 0.1] | [1.5, **1**, 0.5, 0.1] |
| | $\eta_l$ | [0.1, 0.05, **0.01**, 0.005] | [1, **0.5**, 0.1, 0.01] | [1.5, **1**, 0.5, 0.1] | [1, 0.5, **0.3**, 0.1] |
| FEDPROX | $\eta$ | [2, **1.5**, 1, 0.5, 0.1] | [1.5, **1**, 0.1] | [1.5, **1**, 0.5, 0.1] | [1.5, **1**, 0.5, 0.1] |
| | $\eta_l$ | [0.1, 0.05, **0.01**, 0.005] | [1, **0.5**, 0.1, 0.01] | [1.5, **1**, 0.5, 0.1] | [1, 0.5, **0.3**, 0.1] |
| | $\mu$ | [1, 0.1, **0.01**, 0.001] | [1, **0.1**, 0.01, 0.001] | [0.1, 0.01, 0.001, **0.0001**, 0.00001] | [0.1, **0.01**, 0.001, 0.0001] |
| SCAFFOLD | $\eta$ | [1.5, **1**, 0.5, 0.1] | [1.5, **1**, 0.1] | [1.5, **1**, 0.5, 0.1] | [1.5, **1**, 0.5, 0.1] |
| | $\eta_l$ | [0.1, 0.05, **0.01**, 0.005] | [0.5, **0.1**, 0.01] | [1.5, **1**, 0.5, 0.1] | [1, 0.5, **0.3**, 0.1] |
| FEDDYN | $\eta$ | [1.5, **1**, 0.5, 0.1] | [1.5, **1**, 0.1] | [1.5, **1**, 0.5, 0.1] | [1.5, **1**, 0.5, 0.1] |
| | $\eta_l$ | [0.1, 0.05, **0.01**, 0.005] | [0.1, **0.01**, 0.005] | [1.5, **1**, 0.5, 0.1] | [1, 0.5, **0.3**, 0.1] |
| | $\alpha$ | [0.1, 0.01, **0.001**, 0.0001] | [0.1, 0.01, **0.001**, 0.0001] | [0.1, **0.009**, 0.001] | [**0.1**, 0.009, 0.001] |
| ADABEST | $\eta$ | [1.5, **1**, 0.5, 0.1] | [1.5, **1**, 0.5, 0.1] | [1.5, **1**, 0.5, 0.1] | [1.5, **1**, 0.5, 0.1] |
| | $\eta_l$ | [0.1, 0.05, **0.01**, 0.005] | [0.1, 0.05, **0.01**, 0.005] | [1.5, **1**, 0.5, 0.1] | [1, 0.5, **0.3**, 0.1] |
| | $\alpha$ | [0.1, 0.01, **0.001**, 0.0001] | [0.1, 0.01, **0.001**, 0.0001] | [0.1, **0.009**, 0.001] | [**0.1**, 0.009, 0.001] |
| MIME | $\eta$ | [2, **1.5**, 1, 0.5, 0.1] | [2, **1.5**, 1, 0.1] | [1.5, **1**, 0.5, 0.1] | [1.5, **1**, 0.5, 0.1] |
| | $\eta_l$ | [0.1, 0.05, **0.01**, 0.005] | [0.5, **0.1**, 0.01] | [1.5, **1**, 0.5, 0.1] | [1, 0.5, **0.3**, 0.1] |
| FEDAVGM | $\eta$ | [1, 0.5, 0.1, **0.05**, 0.01] | [1, **0.1**, 0.05] | [1, **0.5**, 0.1] | [1.5, **1**, 0.5, 0.1] |
| | $\eta_l$ | [0.5, **0.1**, 0.05, 0.01, 0.005] | [1, **0.5**, 0.1, 0.01] | [1.5, **1**, 0.5, 0.1] | [1, 0.5, **0.3**, 0.1] |
| | $\beta$ | [0.99, 0.9, **0.85**, 0.8] | [0.99, 0.9, **0.85**, 0.8] | [0.99, **0.9**, 0.85] | [0.99, **0.9**, 0.85] |
| FEDACG | $\eta$ | [1, 0.5, 0.1, **0.05**, 0.01] | [1, **0.1**, 0.05] | [0.5, **0.1**, 0.05] | [1.5, **1**, 0.5, 0.1] |
| | $\eta_l$ | [0.5, **0.1**, 0.05, 0.01, 0.005] | [0.5, **0.1**, 0.01] | [1.5, **1**, 0.5, 0.1] | [1, 0.5, **0.3**, 0.1] |
| | $\lambda$ | [0.99, **0.9**, 0.85] | [0.99, **0.9**, 0.85] | [0.99, **0.9**, 0.85] | [0.99, **0.9**, 0.85] |
| | $\beta$ | [0.1, **0.01**, 0.001] | [0.1, **0.01**, 0.001] | [0.1, 0.01, 0.001, **0.0001**, 0.00001] | [0.1, **0.01**, 0.001, 0.0001] |
| MIMEMOM | $\eta$ | [1, 0.5, **0.1**, 0.05] | [1.5, **1**, 0.5, 0.3, 0.1, 0.05] | [1, 0.5, **0.1**, 0.05] | [1.5, **1**, 0.5, 0.1] |
| | $\eta_l$ | [0.1, 0.05, **0.01**, 0.005] | [0.5, 0.1, 0.05, 0.03, **0.01**, 0.005] | [1.5, **1**, 0.5, 0.1] | [1, 0.5, 0.3, **0.1**, 0.05] |
| | $\beta$ | [0.99, 0.95, **0.9**, 0.85, 0.8] | [0.99, 0.95, 0.9, **0.85**, 0.8] | [0.99, **0.9**, 0.85] | [0.99, **0.9**, 0.85] |
| MIMELITEMOM | $\eta$ | [1, 0.5, **0.1**, 0.05] | [1.5, **1**, 0.5, 0.3, 0.1] | [1, 0.5, **0.1**, 0.05] | [1.5, **1**, 0.5, 0.1] |
| | $\eta_l$ | [0.1, 0.05, **0.01**, 0.005] | [0.1, 0.05, 0.03, **0.01**, 0.005] | [1.5, **1**, 0.5, 0.1] | [1, 0.5, 0.3, **0.1**, 0.05] |
| | $\beta$ | [0.99, **0.9**, 0.85, 0.8] | [0.99, 0.95, 0.9, **0.85**, 0.8] | [0.99, **0.9**, 0.85] | [0.99, **0.9**, 0.85] |
| FEDCM | $\eta$ | [1, 0.5, **0.1**, 0.05] | [1.5, **1**, 0.5, 0.1] | [1, 0.5, **0.1**, 0.05] | - |
| | $\eta_l$ | [1, 0.5, **0.1**, 0.05] | [1, 0.5, **0.1**, 0.5] | [1.5, **1**, 0.5, 0.1] | - |
| | $\alpha$ | [0.05, **0.1**, 0.5] | [0.05, **0.1**, 0.5] | [0.05, **0.1**, 0.5] | - |
| **GHBM (ours)** | $\eta$ | [**1**, 0.5, 0.1] | [**1**, 0.1] | [**1**, 0.5, 0.1] | [**1**, 0.5, 0.1] |
| | $\eta_l$ | [0.1, 0.05, **0.01**] | [0.1, **0.01**] | [**1**, 0.5, 0.1] | [1, 0.5, **0.3**, 0.1] |
| | $\beta$ | [**0.9**] | [**0.9**] | [**0.9**] | [**0.9**] |
| | $\tau$ | [5, **10**, 20, 40] | [5, **10**, 20, 40] | [5, **10**, 20, 40] | [5, 10, **20**, 40] |
| **FEDHBM(ours)** | $\eta$ | [**1**, 0.5, 0.1] | [**1**, 0.1] | [**1**, 0.5, 0.1] | [**1**, 0.5, 0.1] |
| | $\eta_l$ | [0.1, 0.05, **0.01**] | [0.1, **0.01**] | [**1**, 0.5, 0.1] | [1, 0.5, **0.3**, 0.1] |
| | $\beta$ | [**1**, 0.99, 0.9] | [**1**, 0.99, 0.9] | [**1**, 0.99, 0.9] | [**1**, 0.99, 0.9] |

## C.4 Hyperparameters

For ease of consultation, we report the hyper-parameters grids as well as the chosen values in Table 7. For GLDV2 and INATURALIST we only test the best SOTA algorithms: FEDAVG and FEDAVGM as baselines, SCAFFOLD and MIMEMOM.

**MOBILENETV2.** For all algorithms we perform $E = 5$ local epochs, and searched $\eta \in \{0.1, 1\}$ and $\eta_l \in \{0.01, 0.1\}$, and found $\eta = 0.1, \eta_l = 0.1$ works best for FEDAVGM, while $\eta = 1, \eta_l = 0.1$ works best for the others. For INATURALIST, we had to enlarge the grid for SCAFFOLD and MIMEMOM: for both we searched $\eta \in \{10^{-3/2}, 10^{-1}, 10^{-1/2}, 1\}$ and $\eta_l \in \{10^{-2}, 10^{-3/2}, 10^{-1}, 10^{-1/2}\}$.

**VIT-B\16.** For all algorithms we perform $E = 5$ local epochs, and searched $\eta \in \{0.1, 1\}$ and $\eta_l \in \{0.03, 0.01\}$ following [Steiner et al., 2022], and found $\eta = 0.1, \eta_l = 0.03$ works best for FEDAVGM, while $\eta = 1, \eta_l = 0.03$ works best for the others.

## C.5 Implementation Details

We implemented all the tested algorithms and training procedures in a single codebase, using PYTORCH 1.10 framework, compiled with CUDA 10.2. The federated learning setup is simulated by using a single node equipped with 11 Intel(R) Core(TM) i7-6850K CPUs and 4 NVIDIA GeForce GTX 1070 GPUs. For the large-scale experiments we used the computing capabilities offered by LEONARDO cluster of CINECA-HPC, employing nodes equipped with 1 CPU Intel(R) Xeon 8358 32 core, 2,6 GHz CPUs and 4 NVIDIA A100 SXM6 64GB (VRAM) GPUs. The simulation always runs in a sequential manner (on a single GPU) the parallel client training and the following aggregation by the central server.

**Practicality of Experiments.** Under the above conditions, a single FEDAVG experiment on CIFAR-100 takes $\approx 02\!:\!05$ hours (CNN, with $T = 20.000$) and $\approx 03\!:\!36$ hours (RESNET-20, with $T =$

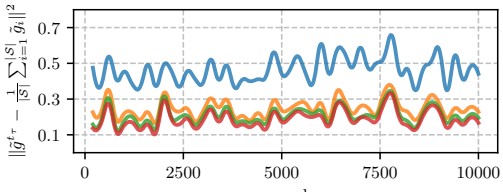 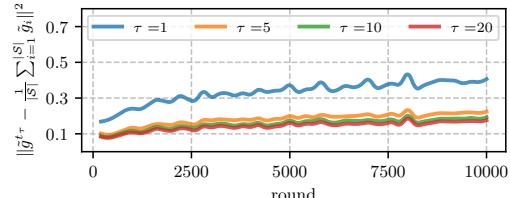

Figure 6: **Reusing old gradients is beneficial, despite the introduced lag.** The plot shows the empirical measure of the deviation between (i) the average of the last $\tau$ server pseudo-gradient (at different parameters) and (ii) the server-pseudo gradient calculated over all the clients (at the same parameters), varying $\tau$, on CIFAR-100 with RESNET-20, in non-iid ($\alpha = 0$, left) and iid ($\alpha = 10.000$, right) settings.

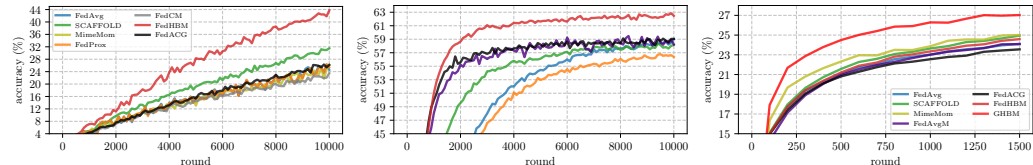

Figure 7: **GHBM largely outperforms state-of-the-art methods:** the plots show the test accuracy (%) over rounds, with RESNET-20 on CIFAR-100, both in NON-IID (left) and IID (middle) settings, and on STACKOVERFLOW (right). GHBM always displays much faster convergence and higher accuracy, even when distributions are IID, confirming robustness w.r.t. heterogeneity and better dependency on stochastic noise.

10.000). For SCAFFOLD we always use the "option II" of their algorithm [Karimireddy et al., 2020] to calculate the client controls, incurring almost no overhead in our simulations. We found that using "option I" usually degrades both final model quality and requires almost double the training time, due to the additional forward+backward passes. Conversely, all MIME's methods incur a significant overhead due to the additional round needed to calculate the full-batch gradients, taking $\approx$ 10:40 hours for CIFAR-100 with RESNET-20. On SHAKESPEARE and STACKOVERFLOW, FEDAVG takes $\approx$ 22 minutes and $\approx$ 3.5 hours to run respectively $T = 250$ and $T = 1500$ rounds.

## C.6 Additional Experiments

**Experiments on CIFAR-10** Table 8 reports the results of experiments analogous to the ones presented in Tab. 1. For the main paper, we report experiments on CIFAR-100, as it is a more complex dataset and often a more reliable testing ground for FL algorithms. Indeed, sometimes algorithms perform well on CIFAR-10 but worse on CIFAR-100 (as for the already discussed case of FEDDYN). Results in Tab. 8 confirm the findings of the main paper: under extreme heterogeneity, some algorithms behave inconsistently across CNN and RESNET-20 (notice that FEDDYN and MIMELITEMOM only with CNN improve FEDAVG. Conversely, LOCALGHBM and FEDHBM both consistently improve the state-of-art by a large margin.

Table 8: **Test accuracy (%) comparison of SOTA FL algorithms in a controlled setting.** Best result is in **bold**, second best is underlined.

| METHOD | CIFAR-10 (RESNET-20) | | CIFAR-10 (CNN) | |
|---|---|---|---|---|
| | NON-IID | IID | NON-IID | IID |
| FEDAVG | $61.0_{\pm 1.0}$ | $86.4_{\pm 0.2}$ | $66.1_{\pm 0.3}$ | $83.1_{\pm 0.3}$ |
| FEDPROX | $61.0_{\pm 1.8}$ | $86.7_{\pm 0.2}$ | $66.1_{\pm 0.3}$ | $83.1_{\pm 0.3}$ |
| SCAFFOLD | $71.8_{\pm 1.7}$ | $86.8_{\pm 0.3}$ | $74.8_{\pm 0.2}$ | $82.9_{\pm 0.2}$ |
| FEDDYN | $60.2_{\pm 3.0}$ | $87.0_{\pm 0.3}$ | $70.9_{\pm 0.2}$ | $83.5_{\pm 0.1}$ |
| ADABEST | $73.6_{\pm 3.0}$ | $86.7_{\pm 0.5}$ | $66.1_{\pm 0.3}$ | $83.1_{\pm 0.4}$ |
| MIME | $53.7_{\pm 2.9}$ | $86.7_{\pm 0.1}$ | $75.1_{\pm 0.5}$ | $83.1_{\pm 0.2}$ |
| FEDAVGM | $66.0_{\pm 2.2}$ | $87.7_{\pm 0.3}$ | $67.6_{\pm 0.3}$ | $83.6_{\pm 0.3}$ |
| FEDCM (GHBM $\tau=1$) | $65.2_{\pm 3.2}$ | $87.1_{\pm 0.3}$ | $69.0_{\pm 0.3}$ | $83.4_{\pm 0.3}$ |
| FEDADC (GHBM $\tau=1$) | $65.7_{\pm 3.0}$ | $87.1_{\pm 0.2}$ | $66.1_{\pm 0.3}$ | $83.4_{\pm 0.3}$ |
| MIMEMOM | $69.2_{\pm 3.6}$ | $88.0_{\pm 0.1}$ | $80.9_{\pm 0.4}$ | $83.1_{\pm 0.2}$ |
| MIMELITEMOM | $57.0_{\pm 0.9}$ | $88.0_{\pm 0.4}$ | $78.8_{\pm 0.4}$ | $83.2_{\pm 0.3}$ |
| **LOCALGHBM (ours)** | $\underline{80.6}_{\pm 0.3}$ | $\underline{88.8}_{\pm 0.1}$ | $\underline{81.1}_{\pm 0.3}$ | $\underline{83.7}_{\pm 0.1}$ |
| **FEDHBM (ours)** | $\mathbf{83.4}_{\pm 0.3}$ | $\mathbf{89.2}_{\pm 0.1}$ | $\mathbf{81.7}_{\pm 0.1}$ | $\mathbf{83.8}_{\pm 0.1}$ |

