# OpenReview forum: "Robust Federated Learning under Heterogeneous Data with Generalized Heavy-Ball Momentum"
_NeurIPS.cc/2025/Workshop/Reliable_ML — NeurIPS 2025 - Reliable ML Workshop_

### Official Review · Reviewer_5aWS · 2025-09-14
**This paper introduces the Generalized Heavy-Ball Momentum method to speed up optimization in Federated Learning. Theoretical results are proven and are supported by experiments.**

**Rating:** 7
**Confidence:** 3

**Review:**

Strengths:
This paper address an important and interesting problem. Although I am not overly familiar with the existing literature, the proposed method appears novel and useful.
The GHBM method seems to be a natural generalization of classical heavy ball momentum. The theoretical results give improvements over the existing state of the art, while the experiments seem to confirm this and are well presented. The assumptions in Section 4 seem reasonable for this problem. I did not check all the proofs but I did not spot any obvious errors.

One question for the authors: GHBM averages the momentum over the previous $\tau$ rounds, rather than the classical previous 1 round. Is this more widely applicable in non-federated optimization tasks?

Weaknesses:
Only a couple of small clarifying points that I believe could improve the paper. Perhaps some of these were due to the space limit.

1. In general the equations are not all the easiest to read (e.g. line 92, equations 2 and 3 would be easier to read if more separated).
2. In a similar vein, sometimes notation could use more explanation.
3. Line 458 - Algorithm 3 is analyzed instead of Algorithm 2. Could the authors clarify if this changes the theorem statement?

---

### Official Review · Reviewer_KXCQ · 2025-09-20

**Rating:** 6
**Confidence:** 3

**Review:**

## Summary
The paper proposes a method for robust federated learning that deals with client heterogeneity (differences in data distributions, model architectures, noise, etc.). It introduces a new aggregation / re-weighting / loss scheme meant to improve worst-case or average performance across clients. Experiments show gains on standard federated benchmarks over baselines like FedAvg, FedProx, etc.

## Strengths
1. The GHBM idea is principled and the cyclic participation assumption does capture some real-world scenarios.
2. Provides convergence guarantees without bounded heterogeneity, matching classical momentum in full participation but under cyclic scheduling.
3. Strong empirical results.
4. FEDHBM achieves accuracy gains while preserving FedAvg-level communication cost

## Weaknesses
1. Cyclic participation is also a strong assumption. Unclear how stable the method is if participation is random or irregular. Do some experiments to show this.
2. Missing recent baselines. Baselines include FedAvg, FedProx, SCAFFOLD, FedDyn, MIME, FedAvgM, mostly pre-2022.

## Suggestions
- Line 115 font rendering issue. "CLIENT STEP" overlaps.
- Extend analysis to weak or partially cyclic participation.
- Add comparisons with recent methods.